# DAF-16/FOXO and HLH-30/TFEB function as combinatorial transcription factors to promote stress resistance and longevity

Xin-Xuan Lin[1,2,3], Ilke Sen[1,2,3], Georges E. Janssens[1], Xin Zhou[1,2], Bryan R. Fonslow[4], Daniel Edgar[1,2], Nicholas Stroustrup [5,6,7], Peter Swoboda [2], John R. Yates 3rd [4], Gary Ruvkun[8,9] & Christian G. Riedel [1,2,3]

The ability to perceive and respond to harmful conditions is crucial for the survival of any organism. The transcription factor DAF-16/FOXO is central to these responses, relaying distress signals into the expression of stress resistance and longevity promoting genes. However, its sufficiency in fulfilling this complex task has remained unclear. Using *C. elegans*, we show that DAF-16 does not function alone but as part of a transcriptional regulatory module, together with the transcription factor HLH-30/TFEB. Under harmful conditions, both transcription factors translocate into the nucleus, where they often form a complex, co-occupy target promoters, and co-regulate many target genes. Interestingly though, their synergy is stimulus-dependent: They rely on each other, functioning in the same pathway, to promote longevity or resistance to oxidative stress, but they elicit heat stress responses independently, and they even oppose each other during dauer formation. We propose that this module of DAF-16 and HLH-30 acts by combinatorial gene regulation to relay distress signals into the expression of specific target gene sets, ensuring optimal survival under each given threat.

[1] Integrated Cardio Metabolic Centre (ICMC), Department of Medicine, Karolinska Institute, Blickagången 6, 14157 Huddinge, Sweden. [2] Department of Biosciences and Nutrition, Karolinska Institute, Blickagången 16, 14157 Huddinge, Sweden. [3] European Research Institute for the Biology of Ageing, University of Groningen, Antonius Deusinglaan, 1, 9713AV Groningen, The Netherlands. [4] Department of Chemical Physiology, The Scripps Research Institute, 10550 North Torrey Pines Road, La Jolla, CA 92037, USA. [5] Centre for Genomic Regulation (CRG), The Barcelona Institute of Science and Technology, C/ Dr. Aiguader, 88, 08003 Barcelona, Spain. [6] Universitat Pompeu Fabra (UPF), C/ Dr. Aiguader, 80, 08003 Barcelona, Spain. [7] Department of Systems Biology, Harvard Medical School, 200 Longwood Ave, Boston, MA 02115, USA. [8] Department of Molecular Biology, Massachusetts General Hospital, 185 Cambridge Street, Boston, MA 02114, USA. [9] Department of Genetics, Harvard Medical School, 77 Avenue Louis Pasteur, Boston, MA 02115, USA. These authors contributed equally: Xin-Xuan Lin, Ilke Sen. Correspondence and requests for materials should be addressed to C.G.R. (email: christian.riedel@ki.se)

I n the wild, organisms are constantly exposed to stresses and privations that put their survival at risk. Sophisticated signaling pathways have evolved that allow organisms to sense these conditions and to respond to them accordingly[1]. Common strategy for most of these pathways is the relay of distress signals into transcriptional changes, in particular the induction of genes that promote stress resistance, slow down the aging process, and infer longevity, which improves the organisms' chances of survival.

Careful coordination of these signaling pathways and their transcriptional outcomes is crucial, so that responses are both effective for their task but also energy efficient, making best use of an organism's resources. This includes that no responses should be triggered which, although they may be helpful under some circumstances, do not provide a benefit under the given threat.

Studies from the last two decades have identified the conserved forkhead transcription factor DAF-16/FOXO as a central player and point of coordination in many of these response pathways[2]. Most importantly, it is the major downstream effector of the nutrient-sensing insulin/IGF signaling (IIS) pathway. Under favorable conditions, IIS is active and leads to the phosphorylation of DAF-16/FOXO by AKT and SGK kinases, resulting in its cytoplasmic sequestration by 14-3-3 proteins away from its target genes. However, under a variety of stressful conditions, e.g. low IIS, but also starvation, infertility, heat, UV, or oxidative stress, this transcription factor is released from 14-3-3 proteins and enters the nucleus to regulate the expression of stress resistance and longevity promoting target genes[1,3].

To date, many upstream signaling pathways have been described to activate DAF-16/FOXO[2], mainly by changing DAF-16/FOXO's posttranslational modification landscape, which leads to its dissociation from 14-3-3 proteins, nuclear entry, and eventually the regulation of stimulus-specific sets of target genes. Given the complexity of the task to relay nature's diverse distress signals into customized responses, the question arises whether DAF-16/FOXO alone is sufficient to fulfill it, or whether there exist other transcription factors with complementary functions that DAF-16/FOXO must synergize with.

Here we address this question and identify the conserved basic helix-loop-helix transcription factor HLH-30 as a second central player in these pathways, operating in close cross-talk with DAF-16/FOXO. HLH-30, as well as its closest human orthologue, Transcription Factor EB (TFEB), have previously been described as starvation-responsive master regulators of lysosome biogenesis and autophagy[4–6], which are important processes in the context of metabolism, aging, and thus the promotion of longevity. In this study, we show that DAF-16/FOXO can form a complex with HLH-30/TFEB and that the two function as combinatorial transcription factors, co-regulating many target genes. Their cooperation and cross-talk ensure customized transcriptional responses to nature's diverse threats, in particular an elaborate control of the organism's stress resistance, certain aspects of development, and its longevity.

## Results

**DAF-16/FOXO and HLH-30/TFEB can form a complex.** In a previous search for binding partners of DAF-16/FOXO, we conducted large-scale purifications of GFP-tagged DAF-16 from whole *C. elegans*, using three different genetic backgrounds: wild type, *daf-2(e1370)* (a conditional mutant of the insulin/IGF receptor gene, which leads to reduction of IIS and thus DAF-16 activation), and *daf-18(mg198)* (a PTEN mutant that leads to constitutively active IIS and thus DAF-16 inactivation)[7]. Subsequent analyses of co-purifying proteins by mass spectrometry identified 133 specific binding partners of DAF-16, several of

which are well established, e.g., the 14-3-3 proteins FTT-2 and PAR-5, two negative regulators of DAF-16[8], and the chromatin remodeling complex SWI/SNF, required for activation of many DAF-16 target genes[7] (see Fig. 1a for the 20 most abundant binding partners of DAF-16). However, the roles of the other binding partners in the context of DAF-16 functions have remained largely elusive. Being interested in other transcription factors that DAF-16 might closely synergize with, we focused on the transcription factor most abundant in DAF-16 purifications: the conserved helix-loop-helix transcription factor HLH-30 (Fig. 1a). Notably, while we found 14-3-3 proteins like FTT-2 most abundant in purifications of inactive DAF-16 (*daf-18* background), HLH-30 was found to co-purify preferentially with DAF-16 in *daf-2* and to a lesser extent wild type backgrounds (Fig. 1b), suggesting that the DAF-16–HLH-30 interaction occurs preferentially when DAF-16 is active and localized in the nucleus. We repeated such large-scale DAF-16 purifications using a different anti-GFP antibody and independently constructed transgenic lines with consistent results (Supplementary Fig. 1a). Next, we validated the DAF-16–HLH-30 interaction under low IIS by co-immunoprecipitation (co-IP) experiments, purifying HLH-30::GFP from *daf-2* mutant animals expressing both HLH-30::GFP and DAF-16::FLAG. Benzonase was added, to exclude any nucleic acid-mediated interactions (Fig. 1c). Finally, we asked if this physical interaction between DAF-16 and HLH-30 is direct, or rather mediated by other *C. elegans* proteins: We conducted in vitro binding assays with recombinant proteins expressed in *E. coli*—again in the presence of DNA and RNA removing enzymes. Purified recombinant $GST::HA_4::DAF-16$ was able to bind recombinant $His_6::myc_6::HLH-30$, indicating that the physical interaction between the two transcription factors is direct (Fig. 1d).

Previous size-exclusion chromatography experiments had shown that in vivo DAF-16 shifts to higher molecular weights and thus increasingly incorporates into larger complexes, when it is activated by low IIS[7]. Given our interaction data from above, we wondered if HLH-30 would behave similarly in such analysis. Using animals co-expressing both DAF-16::FLAG and HLH-30::GFP, we observed that both transcription factors had a broad size distribution, migrating mostly as monomers but also in part as higher molecular weight complexes (Supplementary Fig. 1b). Remarkably, complexes containing DAF-16 or HLH-30 migrated at identical sizes and they showed an identical shift to yet higher molecular weight fractions under low IIS (Fig. 1e, Supplementary Fig. 1b; the additionally shown conditions of heat stress and oxidative stress will only be discussed in a later paragraph). These observations provided yet further support for binding between DAF-16 and HLH-30 and their incorporation into larger complexes, predominantly under DAF-16-activating stimuli like low IIS.

To complete our analysis of DAF-16–HLH-30 complex formation, we wondered if this interaction would be conserved across metazoans, most importantly in human. Thus, we conducted co-IPs between the closest human orthologs of DAF-16 and HLH-30: FOXO1, FOXO3 and TFEB. Pulldown of TFEB led to co-immunoprecipitation of FOXO1 and vice versa (Supplementary Fig. 1c). Surprisingly though, no co-immunoprecipitation between TFEB and FOXO3 was observed (Supplementary Fig. 1c), indicating specificity of TFEB for some human FOXO paralogs over others.

We concluded that DAF-16 and HLH-30 form a complex, preferentially when DAF-16 is activated by specific stimuli (here by low IIS), that this complex formation occurs by direct physical interaction between the two transcription factors, independent of DNA or RNA, and that this complex formation is conserved in human.

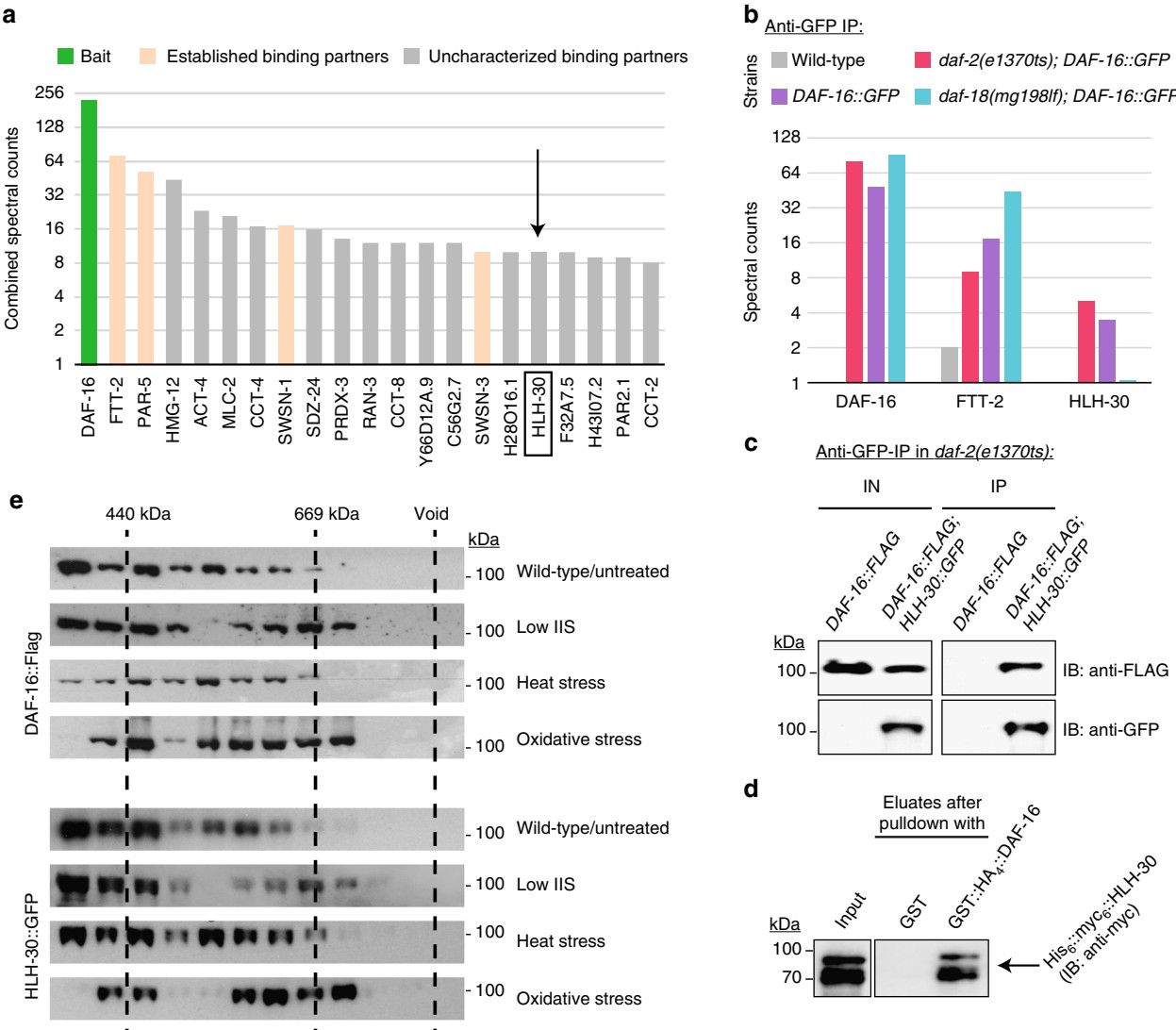

**Fig. 1** DAF-16/FOXO binds to the transcription factor HLH-30/TFEB. **a**, **b** DAF-16 binds to HLH-30, preferentially when it is activated by low IIS. Large-scale anti-GFP immunoprecipitations from whole-worm lysates of animals expressing DAF-16::GFP in either a wild type, *daf-2(e1370ts)*, or *daf-18(mg198lf)* background. Immunoprecipitated material was analyzed by mass spectrometry (LC-MS/MS). In **a**, spectral counts from all three purifications were combined and the 21 most abundant proteins are shown, including the bait DAF-16 and several of its established binding partners. The arrow indicates the most abundant co-purifying transcription factor, HLH-30. In **b**, the spectral counts for each purification were kept separate and are shown for the bait, the 14-3-3 protein FTT-2, as well as HLH-30. **c** Confirmatory co-immunoprecipitation (co-IP). HLH-30::GFP was immunoprecipitated from whole-worm lysates of the indicated *C. elegans* strains using GFP-Trap resin. Benzonase (50 U ml$^{-1}$) was added to eliminate DNA- or RNA-mediated interactions. Inputs (IN) and eluates (IP) of the co-IP were analyzed by SDS-PAGE and western blotting. For the inputs (IN), only fractions were loaded: 5% for the anti-FLAG western blot and 50% for the anti-GFP western blot. IB: antibody used for immunoblot. **d** In vitro binding assay. Recombinantly expressed His$_6$::myc$_6$::HLH-30 was pulled down using Glutathione-Sepharose resin coated either with recombinant GST or GST::HA$_4$::DAF-16. Samples were analyzed by SDS-PAGE and western blotting. For the input, only 50% of the sample were loaded. **e** Size-exclusion chromatography, illustrating the size distributions and thus complex incorporation of DAF-16 and HLH-30 under different conditions. *C. elegans* co-expressing DAF-16::FLAG and HLH-30::GFP were treated in the indicated ways, either by use of *daf-2(e1370ts)* or by exposure for 6 h to either 32 °C or 6 mM t-BOOH. Animals were then immediately frozen, lysed, and their lysates subjected to size-exclusion chromatography on a Superose 6 column. Elution fractions were analyzed by SDS-PAGE and western blotting. Only higher molecular weight fractions are shown. For a full weight-spectrum of untreated animals see Supplementary Fig. 1b

**Both transcription factors enter the nucleus under harmful conditions**. To determine in which tissues, cells, and subcellular compartments the interaction between DAF-16 and HLH-30 might occur, we analyzed the spatial expression patterns of DAF-16::GFP and HLH-30::GFP under their endogenous promoters. Consistent with their interaction, we found that they are globally co-expressed in all tissues and localized diffusely within the cell (Fig. 2a). Those tissues of co-expression include the intestine and neurons (Fig. 2a, b), two of the most relevant

tissues for DAF-16's functions in promoting stress resistance and longevity[9].

As already mentioned, DAF-16/FOXO normally is sequestered in the cytoplasm by 14-3-3 proteins due to AKT/SGK-mediated phosphorylation but can be activated by a variety of stresses, all of which lead to its nuclear translocation and thus engagement in target gene regulation[2]. Recent studies have shown that the activity of HLH-30/TFEB may be regulated in a similar manner, only that the upstream signaling pathways may be somewhat

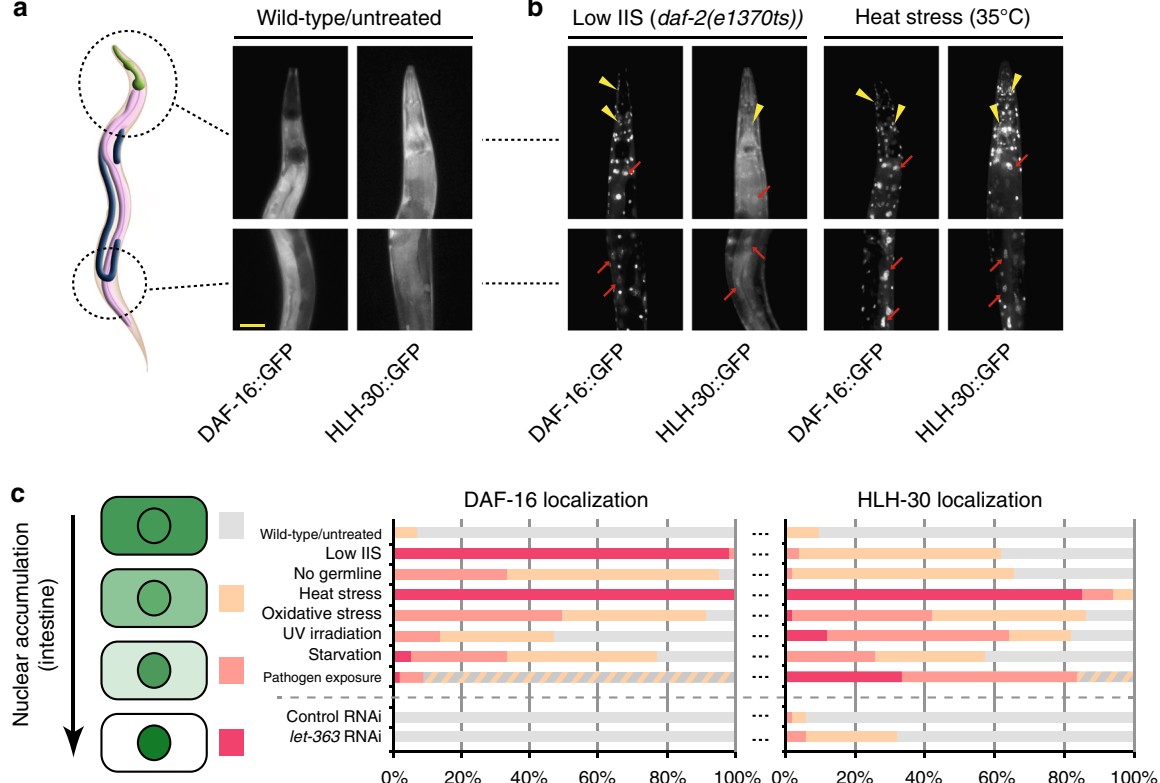

**Fig. 2** DAF-16/FOXO and HLH-30/TFEB are globally expressed and translocate to the nucleus under harmful conditions. **a** DAF-16 and HLH-30 are co-expressed in most tissues of the animal. GFP signal in young adults of the indicated strains is shown. Yellow scale bar: 40 μm. (The anatomical sketch was adapted from wormatlas.org.) **b**, **c** DAF-16 and HLH-30 both translocate into the nucleus upon dire conditions. In **b**, GFP signal in young adults of the indicated strains under low insulin/IGF signaling (IIS, daf-2(e1370ts) for 12 h at 25 °C) or heat stress (1 h at 35 °C) is shown. Yellow arrowheads indicate examples of neuronal and red arrows examples of intestinal nuclear translocation events. In **c**, nuclear translocation of DAF-16 or HLH-30 in day 2 adult animals was scored using DAF-16::GFP or HLH-30::GFP expressing strains under the following stresses and lifespan extending conditions: wild type/untreated, low IIS (daf-2(e1370ts), 12 h at 25 °C), no germline (glp-1(e2141ts), grown from L1 at 25 °C), heat stress (1 h at 35 °C), oxidative stress (12 h on 100 mM tBOOH), UV irradiation (360 mJ cm$^{-2}$ followed by 45 min of recovery), starvation (6 h without food), pathogen exposure (1 day of growth on *Pseudomonas araginosa*; here the categories of no nuclear enrichment (gray) and weak nuclear enrichment (light orange) are combined, since the high background fluorescence of the pathogenic media made distinction of these categories impossible), and reduced TOR signaling (RNAi against *let-363* from the L4 stage). (n = 100)

distinct[10,11]; for example for the mammalian orthologue of HLH-30, TFEB, the predominant kinase to promote sequestration by 14-3-3 proteins is thought to be mTOR (mechanistic target of rapamycin)[11].

Despite such prior knowledge, HLH-30 translocation had not been examined extensively. To fill this gap, we obtained a more comprehensive overview of which stresses can trigger DAF-16 and HLH-30 nuclear translocation and to what extent, focusing on the intestine as an easy-to-score and functionally relevant tissue[9]. We analyzed nuclear accumulation of DAF-16::GFP and HLH-30::GFP in the intestine under a wide variety of stress conditions (Fig. 2c, Supplementary Fig. 2a). Interestingly, DAF-16 and HLH-30 both showed nuclear accumulation under all the tested stimuli. However, their extent of nuclear accumulation differed depending on the stimulus. Under low IIS (daf-2 mutant animals) or absence of a germline (glp-1 mutant animals), DAF-16 translocated more robustly than did HLH-30 (Fig. 2a–c; see also ref. [10]). In contrast, exposure to the pathogen *Pseudomonas aeruginosa* preferentially induced nuclear accumulation of HLH-30 (Fig. 2c; see also ref. [12]). Rather equal levels of nuclear accumulation of both transcription factors were observed for heat stress (32 °C), oxidative stress (6 mM tert-butyl hydroperoxide (t-BOOH)), UV irradiation (360 mJ cm$^{-2}$), and starvation (6 h without food) (Fig. 2a–c). The only stimulus that failed to activate

a transcription factor, namely DAF-16, was RNAi against *let-363*, the gene encoding mTOR in *C. elegans* (Fig. 2c); but we explain this phenotype by the DAF-16::GFP transgene coincidentally used for these experiments. It only expresses the b isoform of DAF-16[13], which is the most commonly studied isoform but known to be refractory to mTOR inhibition[14]. Other DAF-16 isoforms have been shown to get activated by mTOR inhibition[14], suggesting that in fact both DAF-16 and HLH-30, are to some extent responsive to the full range of distress signals that we tested.

Next, we addressed the possibility that DAF-16 nuclear translocation depends on HLH-30 or vice versa. RNAi of *hlh-30* or *daf-16* in DAF-16::GFP or HLH-30::GFP expressing animals, respectively, led to no obvious defects in nuclear translocation of these transcription factors upon heat or oxidative stress (Supplementary Fig. 2b,c), suggesting that DAF-16 and HLH-30 translocate to the nucleus independently.

Taken together, DAF-16 and HLH-30 both responded to an overlapping panel of harmful conditions, which resulted in their nuclear translocation and presumed engagement in the transcriptional regulation of target genes. Notably though, under different stimuli DAF-16 and HLH-30 may translocate to different extents, suggesting that their functions and relevance may differ depending on the physiological context.

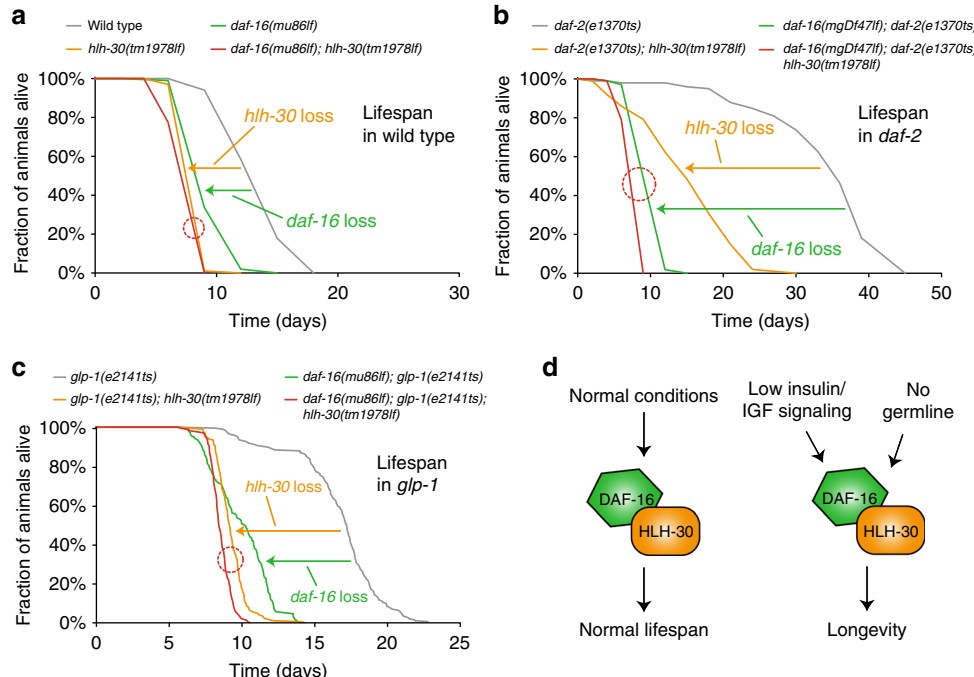

**Fig. 3** DAF-16/FOXO and HLH-30/TFEB require each other to promote longevity. Lifespan phenotypes caused by loss of *daf-16* and/or *hlh-30* in various genetic backgrounds. Red circles indicate the largely absent additive effects caused by joint loss of both transcription factors. **a, b** Animals of indicated genotypes were grown from the L1 stage at 15 °C and then shifted to 25 °C at the L4 stage and their lifespan was monitored. ($n \geq 98$; for detailed statistics including log-rank tests see Supplementary Table 2.) **c** Animals of the indicated genotypes were grown from the L1 stage at 25 °C and their lifespan was monitored. ($n \geq 173$; for detailed statistics including log-rank tests see Supplementary Table 2.) **d** Models illustrating the genetic interaction between *daf-16* and *hlh-30* for the promotion of normal lifespan and longevity

**DAF-16 and HLH-30 require each other to promote longevity.** Having found that both transcription factors can form a complex and that they both translocate into the nucleus under stressful conditions, we then wondered, if they also synergize in their physiological roles. First, we looked at the promotion of a normal lifespan in wild type as well as the promotion of longevity observed in *daf-2(e1370)* or *glp-1(e2141)* mutant animals – all of these being phenotypes that strongly depend on DAF-16[15]. Recent studies have suggested that also HLH-30 influences *C. elegans* lifespan under these conditions[4,10,12]. However, much of this work was based on RNAi methodology, leading to only incomplete loss-of-function phenotypes. Further, a genetic interaction between HLH-30 and DAF-16 had not been explored.

To obtain a genetically more robust view and evaluate a potential genetic interaction between *daf-16* and *hlh-30*, we now used strictly *daf-16* and *hlh-30* null alleles as well as their combination and determined their lifespan phenotypes in various genetic backgrounds. First, we could confirm that not only *daf-16* but also *hlh-30* is required for the normal lifespan of wild type animals and also the longevity of *daf-2* or *glp-1* mutant animals. Remarkably though and in contrast to earlier RNAi experiments, the magnitudes of phenotypes caused by null mutation of *hlh-30* were comparable to those caused by *daf-16* loss, suggesting that HLH-30 is not only a contributor to lifespan phenotypes but actually as relevant as the established key player DAF-16 itself (Fig. 3a–c). Second, we observed that combined loss of both transcription factors had hardly any additive effect (Fig. 3a–c), suggesting that *daf-16* and *hlh-30* are here in a relationship of duplicate recessive epistasis, functioning in the same genetic pathway.

We conclude that DAF-16 and HLH-30 are essential for a normal lifespan of wild type and the longevity of IIS mutant or germline-deficient animals—functions for which they require each other and that they fulfill via the same genetic pathway (Fig. 3d).

**DAF-16 and HLH-30 co-regulate aging-related genes.** Next, we wondered about the mechanism by which DAF-16 and HLH-30 synergize in the promotion of longevity. DAF-16 and HLH-30 are both transcription factors, they both translocate to the nucleus under longevity-promoting conditions, they co-purify, and they function in the same genetic pathway. Thus, we hypothesized that the two transcription factors frequently function in a complex to jointly regulate downstream genes; and it should be via those co-regulated genes that DAF-16 and HLH-30 control the longevity of the organism. To test this hypothesis, we examined the consequences of either *daf-16* or *hlh-30* loss on gene expression, using mRNA sequencing (mRNA-seq) in the three different genetic backgrounds of wild type, *daf-2* mutant, and *glp-1* mutant.

First, we found that both DAF-16 and HLH-30, are required for the correct expression of a large number of genes under each of the tested conditions (Fig. 4c–e). Furthermore, DAF-16 and HLH-30 were both required for much of the lifespan-extending gene expression changes that occur in *daf-2* and in *glp-1* mutants. Of the gene expression changes caused by *daf-2* mutation, loss of *daf-16* fully reverted 80.0% of the activatory and 73.3% of repressive events, while loss of *hlh-30* fully reverted 31.2% and 54.6%, respectively (Fig. 4a). Similarly, of the gene expression changes caused by *glp-1* mutation, loss of *daf-16* fully reverted 11.1% of the activatory and 18.4% of the repressive events, while loss of *hlh-30* fully reverted 7.9% and 12.5%, respectively (Fig. 4b). These analyses showed that not only DAF-16 but also HLH-30 is a key transcription factor driving the gene expression changes in long-lived IIS mutant and germline-deficient animals; and although the reversion phenotypes caused by *hlh-30* loss appear a bit more

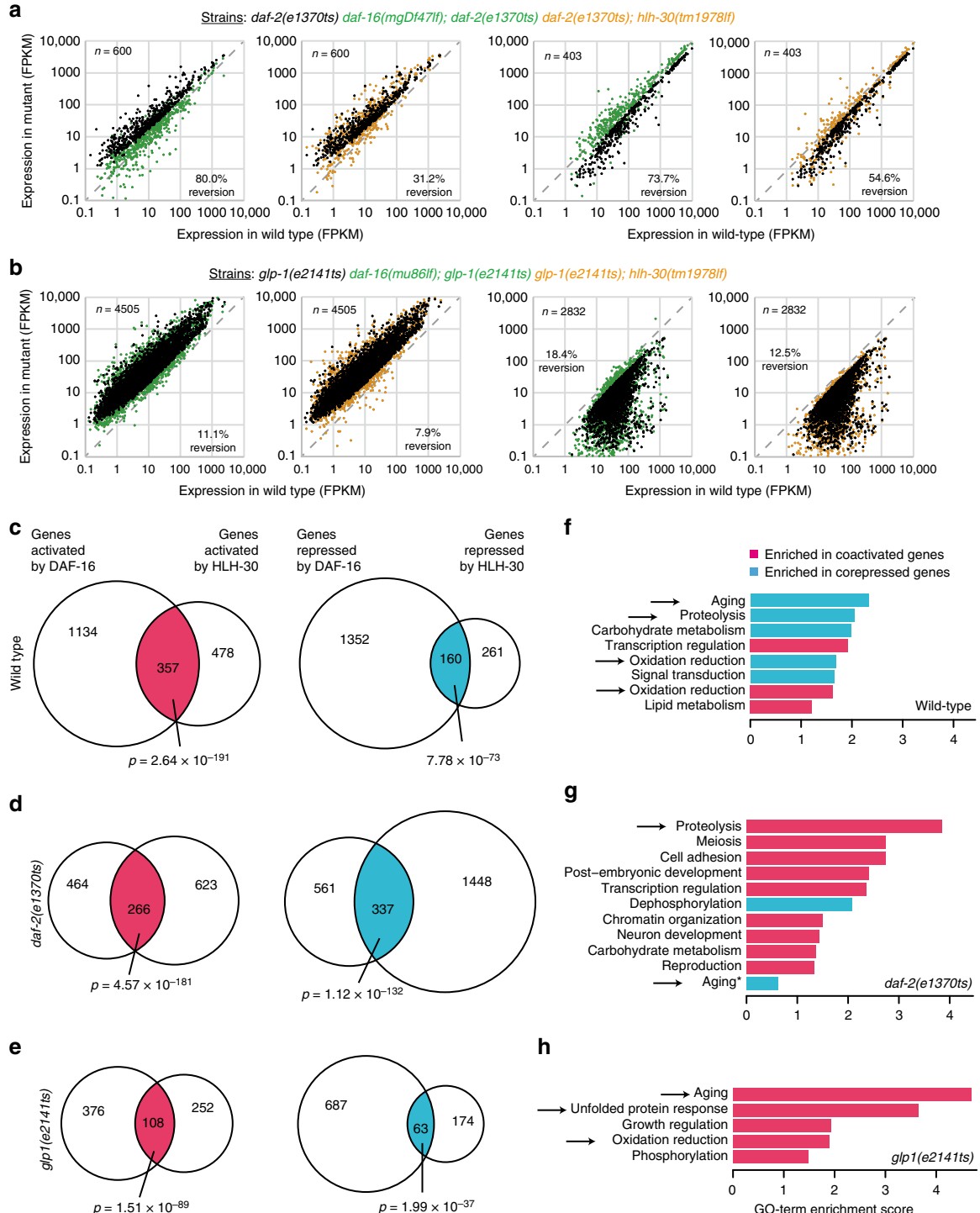

modest, Fig. 3 highlights that the resulting influence of HLH-30 on lifespan is yet comparable to the influence of DAF-16.

Next, we looked for overlaps between the gene sets regulated by DAF-16 and HLH-30: Under all tested conditions, this overlap was substantial (Fig. 4c–e), revealing hundreds of co-regulated target genes. We determined the physiological roles of these co-regulated genes by analyzing them for enrichment of functional classes (Fig. 4f–h). Importantly, GO-terms related to aging, protein homeostasis, and stress resistance were enriched, consistent with our hypothesis that genes co-regulated by DAF-16 and HLH-30 would be crucial for the longevity of the organism. Additionally, GO-terms related to metabolism, growth and development, transcriptional regulation,

and signal transduction emerged from the analysis (Fig. 4f–h). These genes may further contribute to the lifespan phenotypes or indicate other synergistic functions of DAF-16 and HLH-30 beyond the promotion of longevity.

Taken together, our transcriptomic analyses indicate that DAF-16 and HLH-30 are aging-regulatory transcription factors of similar importance that co-regulate a substantial amount of target genes— genes essential for wild type lifespan and the promotion of longevity, in particular under low IIS or in the absence of a germline.

**DAF-16 and HLH-30 colocalize at many promoter regions.** Having shown that DAF-16 and HLH-30 form a complex and

**Fig. 4** DAF-16/FOXO and HLH-30/TFEB co-regulate a large number of genes, in particular genes that influence aging. **a** *C. elegans* with the genotypes wild type, *daf-2(e1370ts)*, *daf-16(mu86lf); daf-2(e1370ts)*, and *daf-2(e1370ts); hlh-30(tm1978lf)* were grown to the L4 stage at 15 °C, then shifted for 12 h to 25 °C, harvested, and their transcriptomes determined by mRNA-seq. The scatter plots on the left show the genes significantly upregulated and the scatter plots on the right the genes significantly downregulated in *daf-2(e1370ts)* compared to wild type animals. **b** *C. elegans* with the genotypes wild type, *glp-1(e2141ts)*, *daf-16(mu86lf); glp-1(e2141ts)*, and *glp-1(e2141ts); hlh-30(tm1978lf)* were grown from the L1 stage at 25 °C, harvested as young adults, and their transcriptomes determined by mRNA-seq. The scatter plots on the left show the genes significantly upregulated and the scatter plots on the right the genes significantly downregulated in *glp-1(e2141ts)* compared to wild type animals. **c** *C. elegans* with the genotypes wild type, *daf-16(mu86lf)*, and *hlh-30 (tm1978lf)* were grown to young adulthood at 20 °C, then harvested, and their transcriptomes determined by mRNA-seq. The Venn diagrams illustrate the number of genes significantly regulated in the mutants compared to wild type as well as their overlap. **d** Based on the data from **a**, these Venn diagrams illustrate the number of genes significantly regulated in the double-mutants compared to *daf-2(e1370ts)* as well as their overlap. **e** Based on the data from **b**, these Venn diagrams illustrate the number of genes significantly regulated in the double-mutants compared to *glp-1(e2141ts)* as well as their overlap. (Significance of gene expression changes in **a–e** was determined by Cuffdiff, using an FDR of 0.05. Significance of gene list overlaps in **c–e** was determined by Fisher's exact test.) **f–h** GO-term enrichment analyses, conducted on the co-activated and co-repressed genes shown in **c–e**. Only GO-terms of significant enrichment are shown (DAVID score $\geq$ 1, *: here the term aging was amongst the 12 most enriched GO-terms, but its score was only 0.62 and therefore below the significance threshold). Arrows highlight any aging-related GO-terms, in particular stress responses, protein homeostasis, and aging itself

that they co-regulate the expression of many genes, we wondered if these transcription factors also co-localize as a complex on chromatin. Thus, we carried out genome-wide mapping of their binding sites by chromatin immunoprecipitation sequencing (ChIP-seq), using *daf-2* mutant animals, as a representative condition where DAF-16 and HLH-30 are active, form a complex, and synergize in the same genetic pathway.

Analyzing this new ChIP-seq data in conjunction with existing data[7] from our lab, we identified 2824 sites bound by DAF-16 and 4932 sites bound by HLH-30. As expected for transcription factors, binding of both DAF-16 and HLH-30, was enriched in promoter regions, mostly within the first 500 bp upstream of the transcriptional start sites (Fig. 5a). We had previously shown that at these promoters DAF-16 functions predominantly as a transcriptional activator[7]. Taking our mRNA-seq data from Fig. 4 into account, we could recapitulate this finding for DAF-16 and made the same observation for HLH-30, namely a significant enrichment of activated but not repressed genes within 2.5 kb downstream of the sites bound by the transcription factor (Fig. 5b).

Next, we asked whether there was significant overlap between DAF-16 and HLH-30 bound sites. Indeed, the overlap was substantial, with more than 41% of the DAF-16 bound sites (1172 sites total) co-occupied by HLH-30 (Fig. 5c, for an example of a co-occupied promoter see the mid panel of Fig. 5e). Furthermore, if DAF-16 and HLH-30 are binding to chromatin as a complex, their spacing in overlapping regions should converge to zero. Plotting distances between summits of DAF-16 bound sites and their closest HLH-30 bound sites, we found that this was the case (Supplementary Fig. 3). We then looked at the identity of the genes immediately downstream of co-occupied sites and observed an enrichment for aging-related functions (Fig. 5d)—consistent with the two transcription factors co-regulating longevity-promoting genes as a complex. Nevertheless, it needs to be noted that 1652 sites were exclusively bound by DAF-16 and 3605 sites were exclusively bound by HLH-30, which shows that while DAF-16 and HLH-30 co-regulate many genes, they have many independent target genes, too.

We then looked for DNA sequence motifs that would be enriched at sites bound by DAF-16 and/or HLH-30: DAF-16 bound sites were enriched for DAF-16-Bound Elements (DBEs, TRTTTAC), while HLH-30 bound sites were enriched for diverse E-boxes (CANNTG), consistent with previous studies[7,16,17]. Sites co-bound by DAF-16 and HLH-30 showed mere combinations of these motifs, with no other apparent sequence features setting them apart (Supplementary Figs. 4, 5). The only noteworthy

observation we made regarding DAF-16-Associated Elements (DAEs, TGATAAG): These elements have been found in the promoters of many DAF-16 regulated genes and are thought to be bound by PQM-1, a transcription factor that assures the baseline expression of DAF-16-regulated genes when DAF-16 is inactive[18]. Here we found that not only sites bound by DAF-16 alone or co-bound by DAF-16 and HLH-30 but also the sites bound by HLH-30 alone were highly enriched for this motif (Supplementary Fig. 4), suggesting that PQM-1 may be involved in assuring the baseline-expression of HLH-30-dependent genes, too. A detailed account of our motif searches can be found in Supplementary Figures 4 and 5 as well as their figure legends.

Given that DAF-16 and HLH-30 form a complex and frequently co-localize on chromatin, we eventually asked, if they exhibit any hierarchy or synergy in their binding to DNA. Therefore, we determined the binding of DAF-16::GFP in *daf-2 (e1370)* and *daf-2(e1370); hlh-30(tm1978)* animals as well as the binding of HLH-30::GFP in *daf-2(e1370)* and *daf-2(e1370); daf-16 (mgDf47)* animals. We observed no significant impact of DAF-16-loss on the binding of HLH-30 to promoter regions co-bound by DAF-16 and HLH-30 (Fig. 5f, $p = 0.59$). However, we observed a small yet significant reduction in DAF-16 binding to these promoter regions in the absence of HLH-30 (Fig. 5f, g, $p = 3.55 \times 10^{-2}$)—a trend not observed when looking across all promoter regions genome-wide (Fig. 5g). This implies that although neither DAF-16 nor HLH-30 are essential for each other's binding to co-bound promoters, HLH-30 may mildly assist DAF-16's binding to such regions.

Taken together, we identified numerous promoter regions directly bound and preferentially activated by DAF-16 or HLH-30. Many of these promoter regions were co-occupied by both transcription factors, with HLH-30 sometimes mildly aiding DAF-16 binding; and the genes downstream of these co-occupied promoters were enriched for aging-related functions. This is consistent with our hypothesis that DAF-16 and HLH-30 promote longevity by both of them getting activated, forming a complex, and this complex binding to promoter regions to drive the expression of longevity-promoting genes.

**Genetic interactions between DAF-16 and HLH-30 are context-dependent.** We have established that DAF-16 tightly cooperates with HLH-30 in the promotion of longevity. However, DAF-16 also has other functions, in particular during stress responses or developmental decisions. A broad role and potential synergy of both transcription factors during stress responses was already indicated by our observation that diverse stresses drive joint

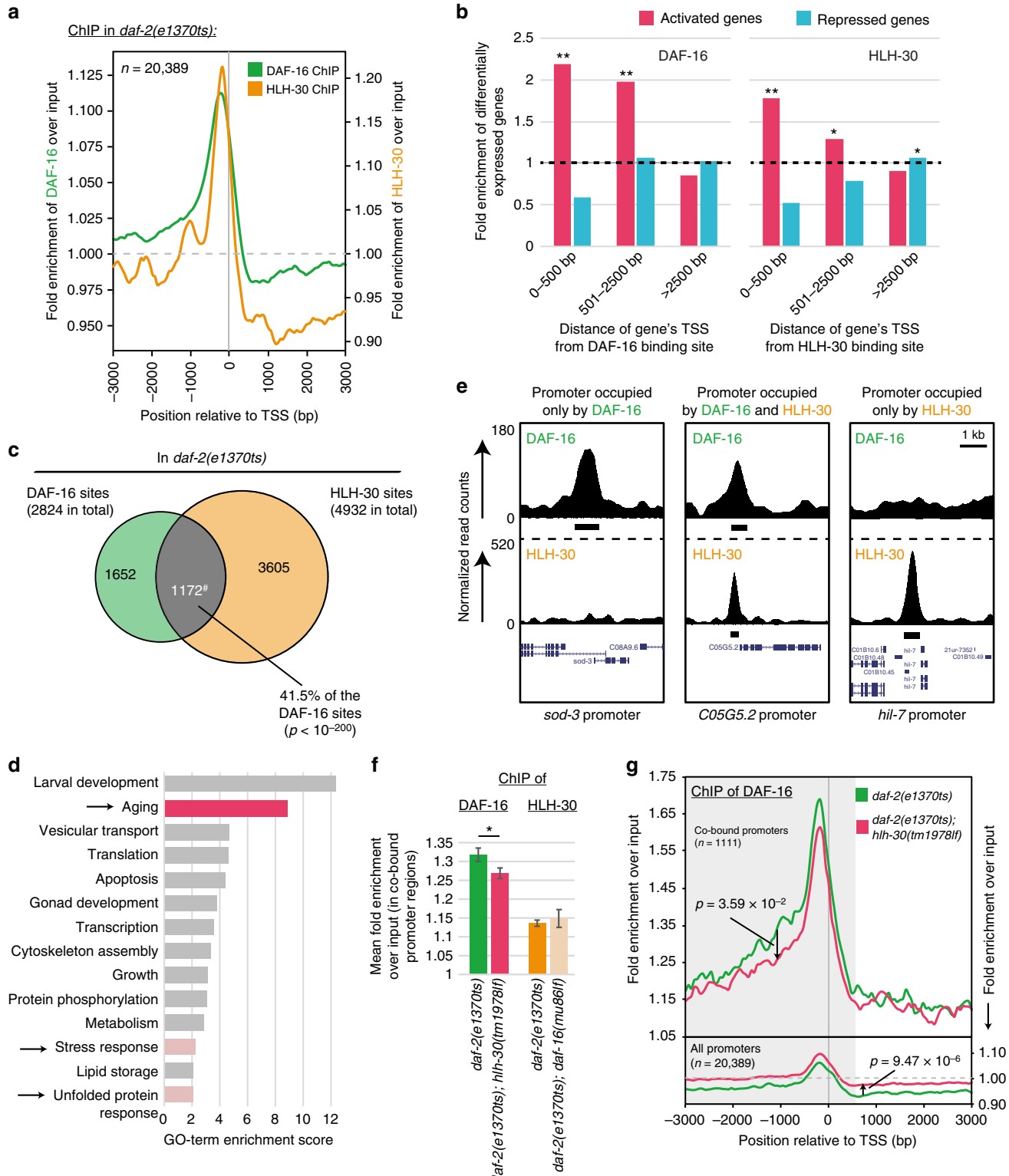

translocation of DAF-16 and HLH-30 into the nucleus (Fig. 2c). Further, longevity is often a result of enhanced stress responses. We thus decided to test the functions of DAF-16 and HLH-30 as well as their genetic interaction in the context of three types of DAF-16-dependent responses, namely the response to oxidative stress, the response to heat stress, and the developmental decision of dauer formation.

First, we determined the survival of wild type as well as *daf-16*, *hlh-30*, or *daf-16*; *hlh-30* mutant animals under oxidative stress or heat stress and found that both transcription factors are important to mediate resistance to these stresses (Fig. 6a, b). Surprisingly though, their genetic interaction differed depending on the stress: While loss of both transcription factors had no additive effect on oxidative stress survival (Fig. 6a), their effect on heat stress survival was completely additive (Fig. 6b), indicating that DAF-16 and HLH-30 function in the same genetic pathway to elicit oxidative stress response but in separate pathways to confer heat stress response.

**Fig. 5** DAF-16/FOXO and HLH-30/TFEB co-occupy many target promoters. **a**–**e** *daf-2(e1370ts)* animals expressing DAF-16::GFP or HLH-30::GFP were grown asynchronously at 15 °C, then shifted for 20 h to 25 °C, until the animals were harvested. Sites bound by DAF-16 and HLH-30 were identified by ChIP-seq using anti-GFP antibody. **a** DAF-16 and HLH-30 bound sites are enriched in the first 500 bp upstream of transcriptional start sites (TSSs). Looking at all genes in the genome ($n = 20{,}389$), enrichment of DAF-16 and HLH-30 in a 6 kb window around their transcriptional start sites (TSSs) is shown. **b** Genes within 2.5 kb downstream of DAF-16 bound sites are enriched for genes transcriptionally activated but not for genes repressed by DAF-16. Similarly, genes within 2.5 kb downstream of HLH-30 bound sites are enriched for genes transcriptionally activated but not for genes repressed by HLH-30. The gene expression information used here was taken from Fig. 4. Significant enrichments are indicated (**: $p < 10^{-8}$, *: $p < 0.01$; hypergeometric test). **c** Venn diagram illustrating the numbers of identified DAF-16 and HLH-30 bound sites as well as their overlap. #: Sometimes a DAF-16 bound site overlapped with multiple HLH-30 bound sites. Thus, the overlap actually comprises 1172 DAF-16 bound sites overlapping with 1327 HLH-30 bound sites. (Significance of site overlap was determined by Fisher's exact test.) **d** GO-term enrichment analysis of the closest genes within 2.5 kb downstream of sites co-bound by DAF-16 and HLH-30. Only GO-terms of significant enrichment are shown (DAVID score ≥ 1). Arrows and pink color highlight GO-terms related to aging. **e** Examples of DAF-16 and HLH-30 binding to different promoter regions in the UCSC genome browser. Black bars indicate binding sites called by MACS. **f, g** *daf-2(e1370ts)* or *daf-2(e1370ts); hlh-30(tm1978lf)* animals expressing DAF-16::GFP as well as *daf-2(e1370ts)* or *daf-2(e1370ts); daf-16 (mu86lf)* animals expressing HLH-30::GFP were grown asynchronously at 15 °C, then shifted for 20 h to 25 °C, until the animals were harvested. Sites bound by DAF-16 or HLH-30 were identified by ChIP-seq using anti-GFP antibody. Differential binding across promoter regions (from −3000 to +600 bp around the TSSs) was evaluated. In **f**, the changes in binding to promoters co-bound by DAF-16 and HLH-30 are shown. (*: $p < 0.05$; $t$-test; error bars indicate variance.) In **g**, binding of DAF-16 and its dependence on *hlh-30* is plotted for a 6 kb window around the TSSs of the indicated promoter regions. ($p$-values were calculated by $t$-test.)

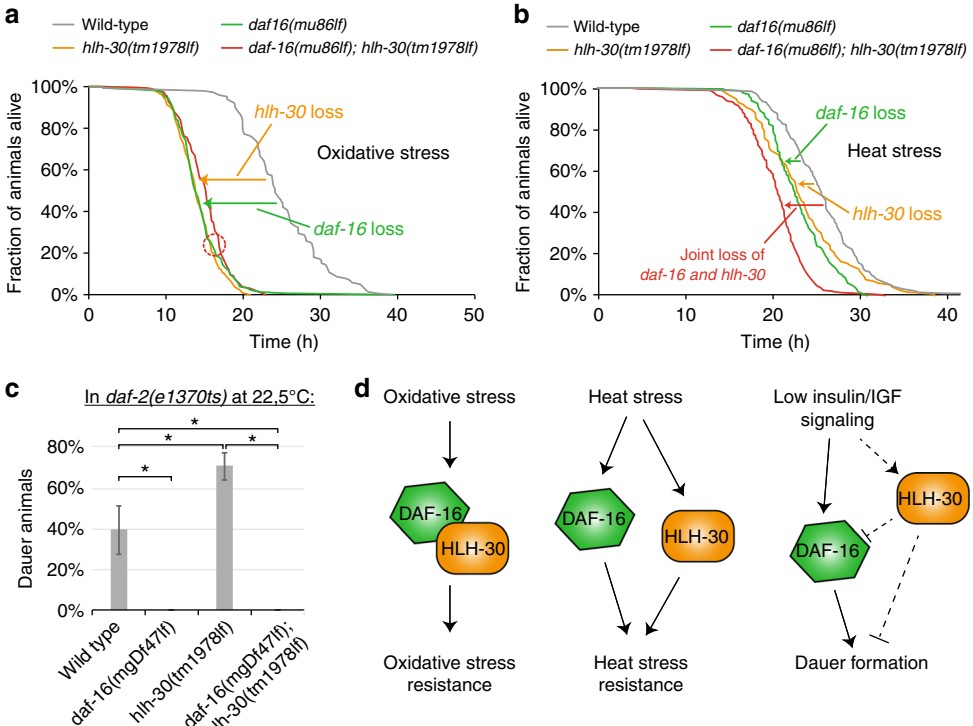

**Fig. 6** The genetic interaction between *daf-16* and *hlh-30* is context-dependent. **a, b** Stress survival phenotypes caused by loss of *daf-16* and/or *hlh-30*. Animals of indicated genotpyes were grown at 20 °C until day 1 adulthood, then transferred to either 6 mM tBOOH (oxidative stress) (**a**) or to 32 °C (heat stress) (**b**). ($n \geq 116$, for detailed statistics including log-rank tests see Supplementary Table 2.) **c** Dauer formation phenotypes caused by loss of *daf-16* and/or *hlh-30* in *daf-2(e1370ts)* animals. Eggs of indicated genotypes were hedged and grown at 22.5 °C for 3 days. Dauer formation was scored based on developmental arrest and morphology. ($n = 100$, results from biological triplicates, *: $p < 0.05$, $t$-test, error bars indicate s.d.) **d** Models illustrating the genetic interactions between *daf-16* and *hlh-30* for the promotion of oxidative stress resistance, heat stress resistance, and dauer formation

Second, we investigated the roles of DAF-16 and HLH-30 during dauer formation: DAF-16 is essential for *C. elegans* to form dauer larvae, a developmental arrest state which allows worms to survive for long periods in harsh environments[19]. To determine whether HLH-30 is also involved in dauer formation, we evaluated *daf-2(e1370)*, *daf-16(mgDf47); daf-2(e1370)*, *daf-2 (e1370); hlh-30(tm1978)*, or *daf-16(mgDf47); daf-2(e1370); hlh-30 (tm1978)* mutant animals at the semi-permissive temperature of 22.5 °C. Under these conditions about 40% of *daf-2(e1370)* animals underwent dauer formation, and loss of *daf-16* fully suppressed this phenotype (Fig. 6c). Surprisingly, loss of *hlh-30* enhanced dauer formation, which indicates that while DAF-16 promotes dauer formation, HLH-30 has a dauer-inhibitory role (Fig. 6c). The enhanced dauer formation in *hlh-30* mutants was suppressed by additional loss of *daf-16*, which supports a greater relevance of DAF-16 for dauer formation but also indicates that in this particular context *hlh-30* may act at least partially upstream of *daf-16*.

We conclude that DAF-16 and HLH-30 are not only required to promote longevity but also to promote stress resistance and

regulate dauer formation. Notably though, these two transcription factors do not always function in the same genetic pathway. Instead, their genetic interaction is stimulus- and thus context-dependent: DAF-16 and HLH-30 function in the same genetic pathway to promote a normal lifespan, longevity, and oxidative stress survival (Figs. 3d, 6d). However, survival of heat stress they promote via separate, parallel pathways (Fig. 6d). Finally, for dauer formation they oppose each other, with HLH-30 moderately preventing it, while DAF-16 strongly promotes it (Fig. 6d).

**DAF-16 and HLH-30 act by stimulus-dependent combinatorial gene regulation**. How can these different context-dependent genetic interactions be explained? To address this question, we initially selected oxidative stress and heat stress as two conditions where the responses rely on both, DAF-16 and HLH-30, but where their genetic interaction differs, with the two transcription factors functioning in the same or separate genetic pathways, respectively. The simplest explanation for these different interactions would be that oxidative stress response is conferred via jointly regulated target genes (similar to the promotion of longevity), while heat stress response is conferred by two separate sets of target genes, one regulated by DAF-16 and one regulated by HLH-30.

To test this hypothesis, we first defined the genes that transcriptionally respond to either oxidative stress or heat stress and thus confer these stress responses: We used wild type animals and exposed them to either control conditions, oxidative stress, or heat stress and determined their transcriptomes by mRNA-seq. This approach identified 957 genes as upregulated and 1214 as downregulated upon oxidative stress (Fig. 7a). 3191 genes were identified as upregulated and 3706 as downregulated upon heat stress (Fig. 7b). Next, we exposed wild type, *daf-16*, *hlh-30*, and *daf-16; hlh-30* mutant animals to the same stress conditions and determined their transcriptomes. For both stresses, this revealed hundreds of genes whose expression depended on DAF-16 or HLH-30, many of which were co-regulated by both transcription factors (Fig. 7c, d, Supplementary Fig. 6). Having shown that DAF-16 and HLH-30 are mostly transcriptional activators (Fig. 5b), we then focused on the genes they activate and within this group of activated genes searched for enrichment of genes we had identified in Fig. 7a, b as induced by oxidative or heat stress in wild type animals. Consistent with our hypothesis, we found that oxidative stress induced genes were particularly enriched amongst genes co-activated by DAF-16 and HLH-30 (Fig. 7c), while heat stress induced genes were particularly enriched amongst genes activated independently, either by DAF-16 or HLH-30 alone (Fig. 7d).

Following this approach, we eventually wanted to understand the distinct relevance and opposing functions of DAF-16 and HLH-30 for dauer formation, too: In Fig. 6c, we had induced dauer formation by partial inactivation of the *daf-2(e1370)* allele at semi-permissive temperature. Thus, we turned to our transcriptomics data in the *daf-2(e1370)* background (shown in Fig. 4a, d) and looked for enrichment of genes involved in dauer formation (276 genes annotated with a GO-term or phenotype related to dauer formation at wormbase.org) amongst the genes regulated by DAF-16 and/or HLH-30 (Fig. 7e). We found that only the genes regulated by DAF-16 alone were significantly enriched for dauer formation-related genes ($p \leq 0.01$), while genes regulated by HLH-30 alone or co-regulated by DAF-16 and HLH-30 were void of significant enrichment. Further, DAF-16 regulated the expression of these genes in a manner that highly correlated with the gene expression changes that occur in dauer forming *daf-2(e1370)* animals, while HLH-30 actively opposed

some of these gene expression changes. A detailed account of this analysis can be found in Supplementary Figure 7 and its legend. We therefore propose that the greater relevance of DAF-16 for dauer formation derives from the dauer program being predominantly regulated by DAF-16 alone, and that HLH-30 mildly opposes dauer formation by counter-regulating a few genes of this program.

Thus, we could show that different roles and context-dependent genetic interactions between DAF-16 and HLH-30 can largely be explained by the distinct distribution of the relevant response genes between genes that are either regulated independently, by DAF-16 or HLH-30 alone, or jointly by both transcription factors. While heat shock response and dauer genes tend to be regulated by either DAF-16 or HLH-30 alone, genes that promote oxidative stress response and longevity must be co-regulated by both transcription factors.

At this point, we reflected back on our IP and size-exclusion chromatography experiments from Fig. 1, demonstrating that binding between DAF-16 and HLH-30 and incorporation into larger complexes is enhanced under low IIS, when both transcription factors are active and function in the same genetic pathway. These results suggested that not only the distribution of target genes amongst individually and co-regulated genes may determine the distinct genetic interactions under different stimuli, but that regulation of DAF-16–HLH-30 complex formation could be a contributing factor, too. Certain upstream stimuli may enhance binding between DAF-16 and HLH-30 and thereby promote the expression of co-regulated target genes. We therefore extended our size-exclusion chromatography analyses of Fig. 1e to include also samples from animals exposed to heat stress and oxidative stress. Here we saw that, just like low IIS, oxidative stress led to a shift of DAF-16 and HLH-30 containing complexes to higher molecular weights, while no such change were observed upon heat stress. This data further supports that conditions requiring the synergy between DAF-16 and HLH-30 in the same genetic pathway promote their complex formation, to activate the expression of co-regulated target genes.

We conclude that the different context-dependent genetic interactions between DAF-16 and HLH-30 can be explained by distinct distribution of stimulus-specific response genes amongst genes individually or co-regulated by these transcription factors, and that this target gene choice may further be influenced by stimulus-specific promotion of the physical DAF-16–HLH-30 interaction.

## Discussion

Efficient stimulus-specific stress responses provide immense advantages to any organism, especially when it must survive in changing environments and under the diverse threats that organisms constantly face in nature. For a long time, DAF-16/FOXO has been considered the most prominent player, right at the center of many of these pathways, relaying distress signals into compensatory transcriptional responses. This is not to say that no other transcription factors have been found involved in these pathways or to synergize with DAF-16/FOXO. Well-known examples would be HSF-1, important for heat stress responses and the longevity of IIS mutants[20,21], SKN-1, important for oxidative stress responses and the longevity of mTOR signaling and IIS mutants[22,23], HIF-1, important for hypoxia-induced longevity[24,25], or the nuclear hormone receptor DAF-12, controlling dauer formation and the longevity resulting from absence of a germline[26,27]. However, all these transcription factors have been more limited than DAF-16/FOXO in their scope, involved in only specific physiological contexts; and when it comes to their synergy with DAF-16/FOXO, none of them have been found to

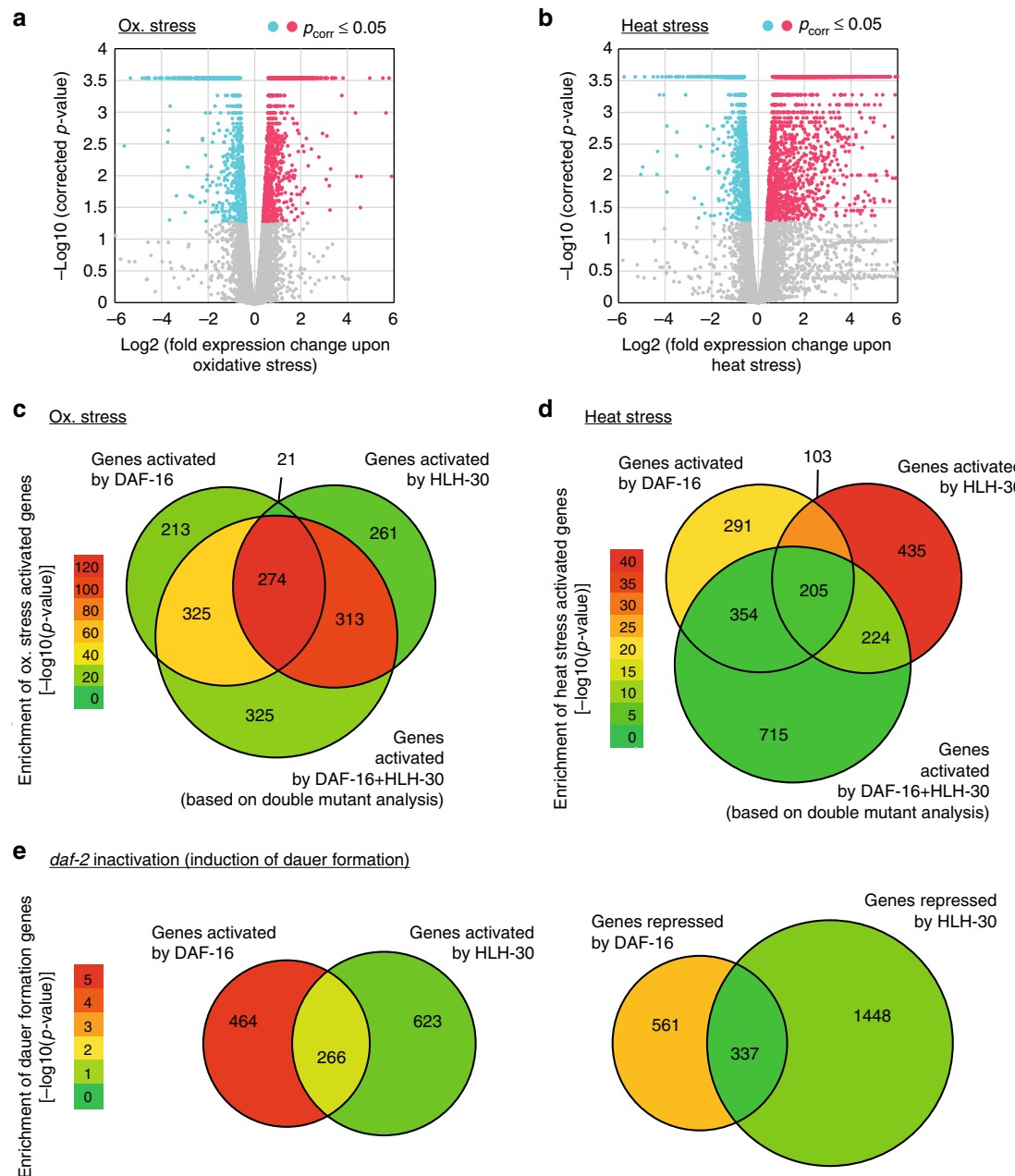

**Fig. 7** The context-dependent genetic interactions between *daf-16* and *hlh-30* can be explained by the distinct distribution of the different response genes between genes individually or co-regulated by these transcription factors. **a**, **b** Genes differentially expressed upon oxidative stress or heat stress. Wild type animals were grown at 20 °C until young adulthood, then transferred for 12 h to either 6 mM tBOOH (oxidative stress) (**a**) or to 32 °C (heat stress) (**b**), harvested, and their transcriptomes determined by mRNA-seq. The volcano plots illustrate the fold expression changes and their significance for each gene of the transcriptome (*n* = 20,389). **c**, **d** *C. elegans* with the genotypes wild type, *daf-16(mu86lf)*, *hlh-30(tm1978lf)*, and *daf-16(mu86lf); hlh-30(tm1978lf)* were grown at 20 °C until young adulthood, then transferred for 12 h to either 6 mM tBOOH (oxidative stress) (**c**) or to 32 °C (heat stress) (**d**), harvested, and their transcriptomes determined by mRNA-seq. The Venn diagrams illustrate the number of genes significantly activated by DAF-16 and/or HLH-30, based on the comparison of their mutants to wild type animals (for Venn diagrams of the repressed genes see Supplementary Fig. 6). False coloring illustrates enrichment for genes responding to oxidative stress (**c**) or heat stress (**d**), as they were determined in panels **a** and **b**, respectively. **e** The same Venn diagrams as in Fig. 4d, but now with false coloring illustrating the enrichment of dauer formation-related genes (*n* = 276, based on GO-term and phenotypic annotations in wormbase.org). (Significance of gene expression changes in **a**–**e** was determined by Cuffdiff, using an FDR of 0.05)

engage in complex formation nor in a broad cooperation with this transcription factor at target promoters in vivo. Thus, the perception of DAF-16/FOXO as a predominant and self-sufficient nexus in the responses to harmful conditions has long remained intact.

Our study now shifts this paradigm, showing that for most purposes DAF-16/FOXO does not function alone. Instead, it partners with the transcription factor HLH-30/TFEB to comprise a sophisticated transcriptional regulatory module (see Fig. 8). This module has the ability to respond to a wide panel of distress signals that converge on either DAF-16, HLH-30, or often both, to regulate target genes important for developmental decisions, stress resistance, and longevity. Some of these target promoters they regulate independently, in particular those regulating dauer

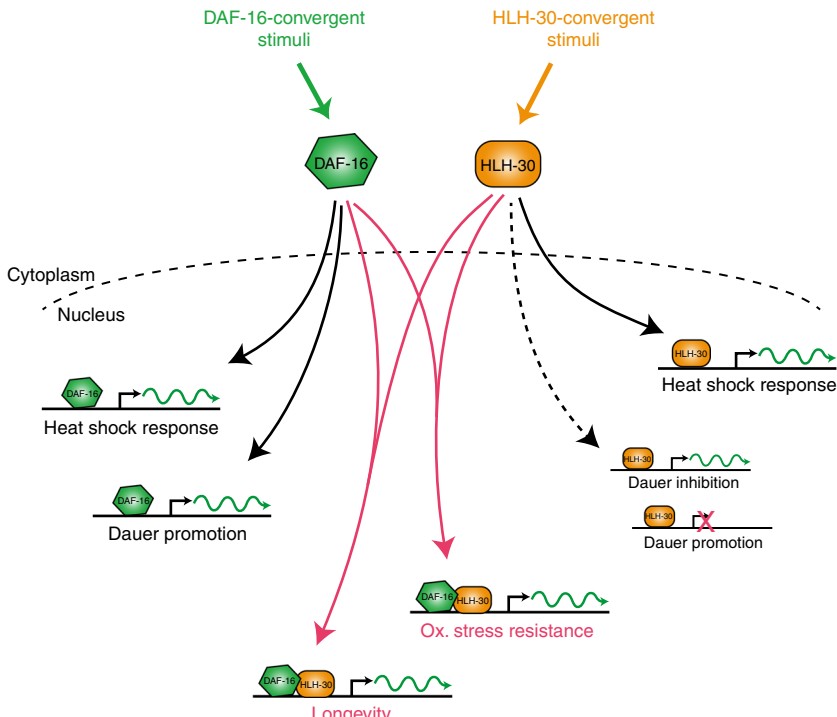

**Fig. 8** Model. DAF-16 and HLH-30 form a sophisticated transcriptional regulatory module. Under non-stressed conditions, both transcription factors reside in the cytoplasm—away from their target genes. However, diverse harmful conditions, some of which converge more on DAF-16, some of which converge more on HLH-30, and some that converge on both, can activate these transcription factors and cause their translocation into the nucleus. Different upstream stimuli and thus different degrees of DAF-16 and/or HLH-30 activation eventually lead to stimulus-specific combinatorial regulation of target genes in the nucleus: Whenever DAF-16 is active, this is sufficient to promote dauer formation and to drive expression of some heat stress response genes. On the other hand, active nuclear HLH-30 is sufficient to increase aspects of heat stress resistance and mildly impair dauer formation. However, a large set of target genes requires the combined action of both transcription factors, where the upstream stimuli must have sufficiently activated and translocated both, DAF-16 and HLH-30, allowing for their complex formation, joint binding to promoter regions, and transcriptional induction at genes that are particularly important for the promotion of oxidative stress resistance and longevity

formation and promoting heat stress resistance. However, regulation of many promoters requires the combinatorial presence of both DAF-16 and HLH-30, presumably as a complex, where they elicit oxidative stress resistance and longevity.

This module of DAF-16 and HLH-30 may comprise the most important regulatory hub known to date for the relay of distress signals into developmental decisions, increased stress resistance, and longevity. It ensures perfectly tailored transcriptional responses to a wide range of stimuli, which we propose arise from the amplitude and balance of DAF-16 and HLH-30 activation, the regulation of their complex formation, as well as the distinct placement of genes intended for the response to different stimuli under the control of promoters regulated either by DAF-16 or by HLH-30 alone, or promoters requiring the combined action of both transcription factors.

For the future, it will be interesting to determine the actual mechanisms by which different stimuli regulate the nuclear entry and complex formation of DAF-16 and HLH-30. Furthermore, Fig. 2c suggests that beyond low IIS, lack of a germline, heat, or oxidative stress, also many other stimuli activate DAF-16 and HLH-30. Investigating the relevance of the DAF-16-HLH-30 module in these other contexts should be very rewarding, too. Finally, DAF-16/FOXO and HLH-30/TFEB are highly conserved across metazoans, regulating similar target genes and physiological processes[2,5,28]. A recent ChIP-seq study in humans showed that DBE and E-box motifs co-occur in many FOXO3-bound promoter regions[29]. And although we observed no physical interaction between human TFEB and FOXO3 in our co-IPs from HEK293T cells, we observed a robust interaction between TFEB

and the FOXO3 paralog FOXO1 (Supplementary Fig. 1c). All of this suggests that combinatorial gene regulation by DAF-16/FOXO and HLH-30/TFEB may be conserved across metazoans. Thus, it will be exciting to explore, if a FOXO–TFEB regulatory module fulfills similar functions in humans, which could lead us to a new and conserved central component of aging regulation and a powerful mechanistic target of interventions against age-related decline and diseases.

## Methods

***C. elegans* strains**. All *C. elegans* strains were grown on *Escherichia coli* OP50 using standard methods[30].

**Strains and alleles**. For a complete list of *C. elegans* strains used in this study, please see Supplementary Table 1.

**RNAi by feeding**. *C. elegans* were grown on *E. coli* HT115 containing dsRNA-expressing plasmids. Clones targeting *let-363/tor* were obtained from[31] and clones targeting *daf-16* and *hlh-30* were obtained from[32]. HT115 containing empty plasmid was used as a control.

**Large-scale immunoprecipitations and mass spectrometry**. Large-scale growth and lysis of *C. elegans*, large scale immunoprecipitations, and eventual analysis of the precipitated material by mass spectrometry for Fig. 1a, b were conducted as described previously[7]. In brief, approximately 20 ml of *C. elegans* pellet were washed into lysis buffer containing 50 mM HEPES at pH7.4, 1 mM EGTA, 1 mM MgCl$_2$, 150 mM KCl, 10% (v/v) glycerol, Complete (Roche), 1 mM phenylmethyl sulphonyl fluoride, and phosphatase inhibitors (Calbiochem) and frozen in liquid nitrogen. Animals were lysed by grinding under liquid nitrogen, NP-40 was added to 0.05% (v/v), and the lysate incubated for 30 min at 4 °C. Finally, the lysate was cleared at 20,000 × g. DAF-16::GFP was immunoprecipitated using anti-GFP antibody (3E6, Invitrogen) coupled to Protein A resin (Biorad).

Immunoprecipitated proteins were eluted using 100 mM glycine at pH2.6. The confirmatory purification in Supplementary Figure 1a was conducted in similar manner (see ref. [33]), but using the anti-GFP antibody mFX73 (Wako, cat # 012-22541).

**Co-immunoprecipitations**. From *C. elegans*: Animals were grown, harvested, and lysed as described previously[7]. Benzonase (50 U ml$^{-1}$) was added to the worm extracts to eliminate DNA- or RNA-mediated interactions. GFP-tagged HLH-30 was immunoprecipitated using GFP-Trap resin (Chromotek) and eluted by boiling in 2× sample buffer. Input (IN) and eluate (IP) samples were analyzed by SDS-PAGE and western blotting. Samples were detected using anti-GFP antibody (Roche, cat # 11814460001) diluted 1:1000 and anti-FLAG antibody (SIG1-25, Sigma, cat # F2555) diluted 1:500 in TBST containing 5% (w/v) milk powder.

From HEK293T cells: HEK293T cells (ATCC, tested to be void of mycoplasma contamination) were grown in advanced Dulbecco's modified Eagle's medium/F12 supplemented with 10% fetal bovine serum and then transfected in 9-cm dishes with 15 μg of plasmid DNA for GFP-FOXO1 (Addgene) or pEGFP-N1-TFEB (Addgene), using the Profection mammalian transfection system (Promega). 12 h post transfection, medium was replaced with fresh antibiotic-free media and LY294002 was added at the final concentration of 20 μM. After 48 h of LY294002 treatment, cells were harvested by scraping and frozen in liquid nitrogen. For the co-IPs, these cell pellets were thawed into lysis buffer (50 mM HEPES at pH7.4, 1 mM EGTA, 1 mM MgCl$_2$, 150 mM KCl, 10% (v/v) glycerol, Complete (Roche), 1 mM phenylmethyl sulphonyl fluoride, phosphatase inhibitors (Roche), and 0.5% (v/v) NP-40) and lysed by shearing through a G30 syringe. Lysates were cleared at 20,000 × g. GFP-tagged proteins were immunoprecipitated using GFP-Trap resin (Chromotek) and eluted by boiling in 2x sample buffer. Input (IN) and eluate (IP) samples were analyzed by SDS-PAGE and western blotting. Samples were detected using the following antibodies: anti-GFP (Roche, cat # 11814460001) diluted 1:1000, anti-FOXO1 (C29H4, Cell Signaling, cat # 2880) diluted 1:500, anti-FOXO3A (Cell Signaling, cat # 9467 s) diluted 1:500, and anti-TFEB (C-6, Santa Cruz, cat # sc-166736) diluted 1:250 in TBST containing 5% (w/v) milk powder.

**Expression of recombinant proteins in *E. coli***. An HA$_4$ tag followed by the cDNA of *daf-16b* was cloned into the GST-fusion expression vector pGEX-4T1, and a myc$_6$ tag followed by the cDNA of *hlh-30a* was cloned into the His$_6$-fusion expression vector pET-28a(+). Empty pGEX-4T1, expressing GST alone, was used as a negative control. All three plasmids were transformed into BL21(DE3)-RIPL (Agilent). Cultures of the resulting bacterial strains were grown to an OD of 0.6-0.7 at 37 °C and then induced with IPTG for 18–20 h at 18 °C. Eventually, the bacteria were harvested and frozen in liquid nitrogen.

**In vitro binding assay**. Bacteria were thawed into PBS buffer containing 1 mM EDTA, 1 mM DTT, 1 mM phenylmethyl sulphonyl fluoride, and Complete (Roche), and incubated with lysozyme for 30 min at 4 °C. Resulting lysates were sonicated, Triton X-100, RNase A, and DNase I were added to concentrations of 1% (v/v), 10 μg ml$^{-1}$, and 5 μg ml$^{-1}$, respectively, and the lysates then incubated for another 30 min at 4 °C. Glutathione-Sepharose 4B resin (GE Healthcare) was added to the lysates containing GST or GST::HA$_4$::DAF-16 proteins and incubated for 2 h at 4 °C. Next, these resins were washed, so that only the purified GST or GST::HA$_4$::DAF-16 proteins would remain, and then incubated with lysate containing His$_6$::myc$_6$::HLH-30 for 2 h at 4 °C. Finally, the resins were transferred to single-use columns (Biorad), washed, and eluted by addition of reduced Glutathione. Eluates were analyzed by SDS-PAGE and western blotting. Proteins were detected using anti-HA antibody (Abcam, cat # ab9110) diluted 1:3000 or anti-myc antibody (9E10, Santa Cruz, cat # sc-40) diluted 1:750 in TBST containing 5% (w/v) milk powder.

**Sample preparation for size-exclusion chromatography**. *C. elegans* were grown asynchronously at 15 °C, then shifted for 20 h to 25 °C. For heat and oxidative stress conditions, worms were additionally subjected to 6 h of either 32 °C or 6 mM tBOOH. Animals were harvested, washed into lysis buffer (50 mM HEPES at pH 7.4, 1 mM EGTA, 1 mM MgCl$_2$, 150 mM KCl, 10% (v/v) glycerol, Complete (Roche), 1 mM phenylmethyl sulphonyl fluoride, and phosphatase inhibitors (Roche)) and lysed by grinding under liquid nitrogen. Lysates were thawed, 0.05% (v/v) NP-40 and Benzonase (50 U ml$^{-1}$) were added, and the lysates were incubated 2 h at 4 °C. Finally, the lysate was cleared at 20,000 × g.

**Size-exclusion chromatography and western blotting**. For size-exclusion chromatography, we used a Superose 6 10/300 column and Äkta Purifier FPLC system (GE Healthcare). First, the column was washed with lysis buffer containing 0.05% (v/v) NP-40 and calibrated using high molecular weight standards (GE Healthcare). Next, the cleared lysates were run on the column. Per lysate, 50 fractions were collected, TCA-precipitated, and eventually analyzed by SDS-PAGE and western blotting. Proteins were detected either by anti-GFP antibody (Roche, cat # 11814460001) diluted 1:1000 or anti-FLAG antibody (SIG1-25, Sigma, cat # F2555) diluted 1:500 in TBST containing 5% (w/v) milk powder.

**Microscopy and nuclear translocation assays**. For imaging, young adult worms were paralyzed by 2,3-butanedione monoxime (BDM), mounted on 2% agarose pads, and imaged using a Zeiss Axio Observer Z1 microscope.

For nuclear translocation scoring, animals were analyzed on day 2 of adulthood after exposure to the following conditions: for low IIS, *daf-2(e1370)* background worms were shifted to 25 °C for 6 h to inactivate IIS; for lack of a germline, *glp-1 (e2141)* background animals were raised at 25 °C from the egg stage to prevent germline proliferation; for heat stress, animals were exposed for 3 h to 35 °C; for oxidative stress, animals were exposed for 3 h to 100 mM tert-butylhydroperoxide (tBOOH); for UV irradiation, animals were exposed to 360 mJ cm$^{-2}$ followed by 45 min of recovery; for starvation, animals were transferred for 6 h to plates without food; for pathogen exposure, animals were grown for 1 day on *Pseudomonas araginosa* PA14. Worms were mounted on 2% agarose pads in the presence of BDM, and the DAF-16::GFP or HLH-30::GFP nuclear translocation in the intestine was scored immediately, using a Zeiss Axio Observer Z1 microscope. To avoid subtle translocation phenotypes caused by exposure to the mounting and imaging conditions, all scoring was conducted within 5 min after mounting.

**Lifespan assays**. The animals were synchronized by egg-laying and grown at 15 °C until the late L4 stage. FUDR was then added to prevent progeny production and the plates were shifted to 25 °C. Survival of animals in all the lifespan assays was measured every 2–3 days, as described previously[34].

Experiments including strains of the *glp-1(e2141)* background were handled with a few differences: the animals were already raised at 25 °C from the embryonic stages to eliminate germline development. Further, survival was recorded and analyzed by a fully automated lifespan machine as previously described[35].

**Stress survival assays**. Animals were synchronized by egg-laying and grown at 15 °C until the L4 stage. Then plates were shifted to 20 °C and FUDR was added to prevent progeny formation. At day 1 of adulthood animals were transferred to plates containing 6 mM tBOOH (oxidative stress) or shifted to 32 °C (heat stress). Their survival was recorded and analyzed by a fully automated lifespan machine as previously described[35].

**Dauer assays**. *C. elegans* strains were synchronized by egg laying and grown for 4 days at 22.5 °C (a semi-permissive temperature for *daf-2(e1370)*). Dauer formation was determined based on developmental arrest and morphology. Results were derived from three biological replicate experiments.

**mRNA isolation and library construction**. Approximately 100 worms were synchronized by egg-laying and raised according to the following conditions: In experiments investigating *daf-2(e1370)* animals, the animals were grown at 15 °C until the L4 stage, then FUDR was added and the plates were shifted to 25 °C for 12 h. In experiments investigating *glp-1(e2141)* animals, the animals were grown at 25 °C until young adulthood. In experiments investigating oxidative or heat stress responses, the animals were grown at 20 °C until the L4 stage, when FUDR was added. From day 2 of adulthood, they were subjected for 6 h to either 6 mM tBOOH or shifted to 32 °C. Some animals were kept at 20 °C in the absence of any stress as controls.

Eventually, all animals were collected, briefly washed, and immediately frozen in liquid nitrogen. Total RNA was extracted using Trizol. mRNA-seq libraries were constructed using a TruSeq RNA SamplePrep V2 kit (Illumina).

**Chromatin immunoprecipitation and library construction**. Animals were grown asynchronously at 15 °C and then shifted to 25 °C for 20 h. ChIP was performed as previously described[36]. In brief, 4–5 ml of frozen ground worm powder were thawed and crosslinked at 20 °C, using 1% (w/v) formaldehyde in PBS. Samples were sonicated and cleared at 20,000 × g. GFP-tagged proteins of interest were immunoprecipitated using polyclonal anti-GFP antibody (Clonetech, cat # 632592) bound to Protein A Dynabeads (Invitrogen). Protein–DNA complexes were then eluted using 1% (w/v) SDS and treated with RNase A and proteinase K. Input and eluate samples were purified using a ChIP-DNA clean up kit (Zymo research). ChIP- seq libraries of input and eluate samples were constructed according to[37]. In brief, ChIP-DNA was end-repaired, A-tailed, ligated to universal adapters, and amplified for 12–15 cycles with indexed primers. Prior to sequencing, excess adapters or large fragments were removed using AMPure XP magnetic beads (Beckman Coulter).

**High-throughput sequencing**. Multiplexed single-end sequencing of mRNA-seq and ChIP-seq libraries were conducted for 50 cycles on an Illumina HiSeq 2000 according to the manufacturer's instructions. Image analysis, base calling and quality scoring were performed in real time with the standard Illumina analysis pipeline using a phiX control.

**Analysis of mRNA-seq data**. Reads were aligned to the *C. elegans* genome with the TopHat (v2.0.8b) software package[38], using known gene model annotations (WS220) and the following parameters: --library-type fr-unstranded --b2-very-sensitive --min-coverage-intron 10 --min-segment-intron 10 --microexon-search --no-

novel-juncs. Transcript abundance (FPKM, fragments per kilobase of transcript per million fragments) and differential expression were calculated using Cuffdiff (v2.1.1) included in the Cufflinks software package[39] using the following parameters: -u --FDR 0.05 --upper-quartile-norm --compatible-hits-norm --library-type fr-unstranded. Conditions of Figs. 4a, b, d, e, g, h and 7e, and Supplementary Fig. 7 were analyzed in biological duplicate. Conditions of Figs. 4c, f, 7a–d and Supplementary Fig. 6 were analyzed in biological triplicate. Replicates were individually mapped and then combined by Cuffdiff. All analyses were limited to protein-coding genes. Statistically significant differentially expressed genes (DEGs) were identified using a 5% FDR. Differential gene expression values were calculated as the ratio of FPKM values. To test for significant overlap between gene lists (Fig. 4c–e, and also the gene enrichment analyses in Fig. 7c–e) the Fisher's exact test was used (fisher.test R function[40]).

**Analysis of ChIP-seq data**. Reads were aligned to the *C. elegans* genome (WS220) using Bowtie (v2.1) with the following parameter: -q. Uniquely mapping reads containing no more than one mismatch were used for peak calling and read density calculations. Enriched peaks were identified using MACS (v2) with the following parameters: --mfold 5,30 --bw 200 --keep-dup auto -q 0.05. Only statistically significant peaks ($q < 1 \times 10^{-6}$) were kept. Genomic regions that are commonly identified in ChIP-seq experiments, so-called hotspots, represent potential artefacts and were removed from the peak data set as previously described[41]. To calculate read densities across transcriptional start sites, the R package ngs.plot was used with the following parameter: -FL 200[42]. To display genome-wide read densities in the UCSC genome browser (Fig. 5e), reads first were extended to mean fragment size (200 base pairs (bp)) in the 3′-direction of the read to more precisely reflect the true binding position, then they were converted to coverage data by the genomecov command of the BEDTools Utility suite[43], and finally they were uploaded to the UCSC genome browser. A smoothing window of 10 pixels was applied.

Overlaps between sets of MACS peaks in Fig. 5c (peaks being referred to as bound sites in the manuscript) were determined using intersectBED and the significance of these overlaps calculated by Fisher's exact test, both being part of the BEDTools Utility suite[43].

Distances between MACS summits in Supplementary Fig. 3 were determined using closestBED of the BEDTools Utility suite[43]. The trendline is based on the equation $y = a \ln(x) + b$ and was fitted in MS Excel.

For *p*-value calculations in Fig. 5f, g, indicating whether DAF-16 binding to promoter regions had significantly changed between the compared conditions, we first calculated read densities in 60 bp bins using ngs.plot (with the parameter -FL 200)[42], covering a region from −2000 to +300 bp across transcriptional start sites. Next, we compared the number of reads in these bins between conditions using *t*-test[44].

Results shown in Fig. 5 and Supplementary Figs. 3 to 5 are based on individual experiments. However, the data has been validated by independent replicate experiments showing highly correlated results (Person correlations of higher than 0.92, Supplementary Fig. 8). Pearson correlations between replicate experiments were determined by BAM file comparisons using multiBamSummary with parameters --minMappingQuality 30 --binSize 1000 and plotCorrelation of the DeepTools software package (v. 2.5.0)[45].

**Associating ChIP-seq peaks with proximal genes**. For each protein-coding gene the closest distance between any of its TSSs and a MACS peak summit was calculated. Only peak summits positioned upstream or within 200 bp downstream of the TSS were considered.

**Identification of enriched DNA motifs within ChIP peaks**. Transcription factor bound sites obtained from ChIP-seq analysis (MACS peaks) were searched for enriched DNA motifs using peak-motifs at http://metazoa.rsat.eu/[46]. Searches were conducted on regions of equal size (500 bp regions surrounding each peak summit) using default parameters and an oligomer length of 6. Randomized sequences of equal length were used as controls. Significance scores were determined as previously described[46].

**Gene functional enrichment analysis and annotation**. Gene functional enrichments were determined by using the DAVID Bioinformatics Resources version 6.7[47]. Annotation clusters determined by DAVID (groupings of related annotation terms) having an enrichment score of ≥ 1 were considered significant and a representative naming for the cluster was derived from the contained Gene Ontology (GO) terms.

**Statistics and reproducibility**. All results presented in this manuscript have been reproduced at least once in independent experiments. Lifespan and stress resistance assays were evaluated by Kaplan–Meier and log-rank tests. The detailed results, also of the replicate experiments, are shown in Supplementary Table 2. Dauer formation was evaluated by Student's *t*-test. *p* values of Fig. 5b were determined by hypergeometric test (phyper R function[40]). Statistical analyses for other experiments are stated in the respective methods sections, tables, or figure legends. Analyses were performed using either SPSS (IBM), Origin (Originlab), MS Excel,

or R[40]. Non-cropped blots of the key electrophoresis data is shown in Supplementary Figs. 9, 10.

## Data availability

The high-throughput sequencing data generated and/or analyzed during this study are available from the authors upon reasonable request as well as from the Sequence Read Archive at NCBI at the following accession codes: SRP152334 and SRP017927. The mass spectrometry data generated and/or analyzed during this study are available from the authors upon reasonable request as well as from the PeptideAtlas at the following accession codes: PASS01224 and PASS00130.

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

## Acknowledgements

We thank Orane Visvikis, Javier Irazoqui, Louis Lapierre, Malene Hansen, James Moresco, Cynthia Kenyon, Shohei Mitani, Alexander Soukas, Eyleen O'Rourke, and Andrew Dillin as well as his lab for materials and advice. We thank Tales Rocha de Moura, Tim Schulte, and Adnane Achour for help with size-exclusion chromatography. We thank Bora Baskaner, Simone Brandenburg, the Lansdorp lab at ERIBA, and the BEA sequencing core at Karolinska Institute for technical support. We thank Jérôme Salignon for help with bioinformatic analyses. Finally, we thank Maria Eriksson and members of the Riedel lab for advice and comments on the manuscript. N.S. was supported by the Spanish Ministry of Economy, Industry and Competitiveness (MEIC) to the EMBL partnership, by the Centro de Excelencia Severo Ochoa, and by the CERCA Programme/Generalitat de Catalunya through an award from the Glenn Foundation for Medical Research. J.R.Y. was supported by the grant NIH P41 GM103533. C.G.R. was supported by the grants VR 2015-03740 and GENiE COST BM1408.

## Author contributions

X.X.L., I.S., X.Z., and C.G.R. conceived and designed the experiments. X.X.L., I.S., G.E.J., X.Z., D.E., and C.G.R. conducted the experiments and/or analyzed the resulting data. B.R. F. and J.R.Y. provided mass spectrometry data and advice, N.S. helped with the setup of the lifespan machine. P.S. and G.R. provided materials and advice. X.X.L., G.E.J., and C. G.R. wrote the manuscript.

## Additional information

**Competing interests:** The authors declare no competing interests.

