## [Peer Review File · Nature Communications]

Reviewers' comments:

Reviewer #1 (Remarks to the Author):

In this work the authors start with a pulldown of DAF-16::GFP to identify binding partners (co-IP and mass spec) and in addition to known partners such as FTT-2, PAR-5, and SWI/SNF. HLH-30 bound to DAF-16 in *daf-2* background (activated DAF-16, nuclear), and this was validated; results suggest that the interaction is not mediated by RNA or DNA (chromatin). HLH-30/TFEB is a well-known transcription factor that has been already described to be involved in lifespan regulation and autophagy.

The translocation results suggest that the two TFs act independently under a wide range of conditions, eliminating binding to one another for the purpose of translocation as a mechanism. Loss of *daf-16* and *hlh-30* genetically suggests the two TFs are both independently required for *daf-2* and *glp-1* lifespan extension. (I'm not sure how "epistatic" is being used in the text, as no order is established when both *lof* mutations lack an additive effect.)

Transcriptional analysis shows that many of DAF-16's targets are also HLH-30 targets, and chIP-seq experiments suggest that HLH-30 activates expression rather than repressing. Additionally, motif analyses suggest that simple promoter binding of the two TFs to their respective motifs (DBE and E-box) regulate binding, rather than the emergence of a new motif. Additionally, the binding of PQM-1 via the DAE may be involved in HLH-30 "baseline" expression. Dauer formation is surprisingly increased by loss of *hlh-30*, opposite to *daf-16*.

-Tissue localization: if the tissue-specific DAF-16 expression lines (Libina, et al 2003; Zhang et al 2013) are used for co-IP/MS (intestine, neurons, muscle, hypodermis) is HLH-30 still pulled down in all of them? A more direct method of interaction should be used to address the tissue-specific interaction between HLH-30 and DAF-16, rather than relying on GFP results.

-Which tissue was used to score nuclear translocation?

-The translocation experiments, genetic evidence, transcription results, and chIP-seq data all suggest that the two TFs act independently. What is the evidence for this sentence?: "This is consistent with our hypothesis that DAF-16 and HLH-30 promote longevity by both of them getting activated, forming a complex, and jointly binding to promoter regions, where they promote expression of downstream genes important to prevent aging and thus extend the lifespan of the animal." or "We have established that DAF-16 tightly cooperates with HLH-30 in the promotion of longevity."

How would their model be different if HLH-30 had simply been characterized without co-IP data? No cooperation is established by any of the data presented to this point in the paper. The transcriptional data and chIP-seq data do not point to the formation of a complex, but rather the independent regulation of two stress-sensitive TFs. Cooperation would be shown by synergetic interactions, which is not shown here; instead, all of these data besides co-IP paint a picture of two ships passing in the night in response to the same beacon but totally unaware of one another. The dauer results further support this view, as DAF-16 and HLH-30 have opposite roles.

The authors discuss in the suggestion that these data shift the paradigm of the field, but there is no data shown here that support the notion that these two TFs form a transcriptional regulatory module; the data suggest that HLH-30 regulates a subset of DAF-16's targets--independently of DAF-16--, that it is often found on the promoters of DAF-16 target genes, and that most (but not all) of the time, HLH-30 also responds in the same way to environmental conditions as DAF-16 does.

How would the authors distinguish their model from one in which two TFs that function independently but are nuclear localized under similar conditions end up near each other on promoters under stress conditions, but do not directly or synergetically act, but instead are simply

more often near each other, so under cross-linking conditions they are more likely to bind to one another due to proximity rather than a real complex formation? It seems that this is in fact the more likely situation; otherwise the stoichiometry should be 1:1, and there should be evidence of cooperatively rather than independence.

Overall, this manuscript does a good job of describing all of the myriad ways that HLH-30 acts similarly to DAF-16 and presents a great deal of information about HLH-30 activity, but the main point, that it binds to DAF-16, remains almost entirely unexplored beyond the initial co-IP results; the nature of this physical interaction that initiated the study is not rigorously explored. While that is not critical for the basic description of HLH-30's targets under various conditions (which itself is valuable information), the use of words like "cooperative" and "complex" to describe its interactions are not justified by the data shown here.

Reviewer #2 (Remarks to the Author):

The Lin et al manuscript reports an interesting collaboration between two transcription factors DAF-16 and HLH-30 in longevity and stress response. Whereas both DAF-16 and HLH-30 have been shown to be important for longevity, their working relationship was unknown. The authors revealed a partnership between the two factors via IP mass spec. Using further genetic and genomic analyses, they presented data to support that DAF-16 and HLH-30 form a protein complex and are recruited to the common set of promoters to regulate their gene expression. More interestingly, under some conditions, DAF-16 and HLH-30 each have their own regulatory targets and thus different functional outcomes. Therefore, DAF-16 and HLH-30 both act together and separately to regulate various physiological outputs.

In general, the data presented are of good quality and largely support the conclusions.

A few comments:

For Fig. 1, representative western blot and nuclear localization results should be shown. The bar graphs are effective for summarizing the results, but actual images that represent the different results will give a better idea about what the results actually look like.

In the Method section, it was stated that all experiments were done at least as duplicates. However, based on the data presented, it is difficult to know how consistent the replicates are. For lifespan data, the quantitative data of the individual trials should be presented as supplementary data.

For the RNA-seq and ChIP-seq data, the overall correlation of the data between the replicates should be shown, e.g. via a simple scatter plot or correlation or PCA analysis. Based on the Method, it is also not clear how the duplicate data are processed. Are the data simply pooled, or are the data analyzed as separate replicates and does the data processing pipeline take the replicates into account and only report consistent peaks?

Letter to the referees:

To begin, we would like to thank the reviewers for their efforts in assessing our manuscript and for their constructive feedback. We appreciate that both reviewers acknowledged the general quality and value of our work, and we were also happy to see that despite their various comments, both reviewers didn't share any specific concerns, which we think illustrates that our original manuscript was void of major shortcomings, or at least shortcomings that would be apparent to every reader.

Following the feedback from reviewer #1 and detailed discussions with the editor, we conducted a number of new experiments to strengthen our manuscript, in particular providing additional biochemical support for the interaction and cooperation between DAF-16 and HLH-30. These experiments are summarized in the following:

- 1) We asked, whether the physical interaction between DAF-16 and HLH-30 is direct and not mediated by other *C. elegans* proteins or DNA/RNA. To address this, we conducted in vitro binding assays in the presence of DNase I and RNase A, using recombinant GST::DAF-16 and His6::HLH-30 expressed in *E. coli*.
→ We found that the physical interaction between DAF-16 and HLH-30 can be recapitulated in vitro and thus is direct (Fig. 1d).
- 2) We asked, if the complex formation between DAF-16/FOXO and HLH-30/TFEB is also conserved in humans. To address this, we conducted co-IPs between human TFEB, FOXO1, and FOXO3 in HEK293T cells under low insulin/IGF signaling (IIS).
→ We found that TFEB can bind to FOXO transcription factors also in humans. Interestingly though, TFEB specifically binds to FOXO1 and not to FOXO3 (Fig. S1c). Overall, this data indicates a possible conservation of our findings in humans. Although future studies will have to validate this further and will also have to address the reasons behind TFEB showing binding specificity to some FOXOs over others.
- 3) Given that DAF-16 and HLH-30 form complexes, we were curious whether DAF-16 and HLH-30 would co-migrate also in size-exclusion chromatography experiments and how their migration would change under different stimuli, i.e. low IIS, oxidative stress, or heat stress. Notably, we had shown in our original manuscript that the binding of DAF-16 to HLH-30 is enhanced under low IIS, and a previous study had shown that DAF-16 shifts to higher molecular weight and thus incorporates into larger complexes under such conditions¹.
→ We found that DAF-16 and HLH-30 co-migrate in size-exclusion chromatography experiments and that both transcription factors shift in their distribution to the same higher-molecular weight fractions, specifically upon stimuli that lead to actions of DAF-16 and HLH-30 in the same genetic pathway, namely under low IIS and oxidative stress (Fig. 1e, S1b). No such shift was observed upon heat stress (Fig. 1e), for which we have shown in Fig. 7d that DAF-16 and HLH-30 respond to it independently, via separate pathways.
This data is consistent with DAF-16 and HLH-30 binding to each other and incorporating into higher-molecular-weight complexes – complexes that form preferentially under stimuli that require the actions of DAF-16 and HLH-30 in the same genetic pathway. Furthermore, this data implicates that the formation of DAF-16–HLH-30 complexes may actually be regulated by upstream stimuli and thus could be part of the mechanism by which certain stimuli promote the expression of DAF-16–HLH-30 co-regulated target genes. Future studies will have to explore this further.
- 4) We asked, if DAF-16 and HLH-30 would show any synergies in promoter binding. To address this, we conducted ChIP-Seq experiments, looking for changes in DNA binding of either DAF-16 in null mutants of *hlh-30* or of HLH-30 in null mutants of *daf-16*.
→ We found no significant change in HLH-30 binding to promoters co-bound by DAF-16, when DAF-16 was missing. However, we observed a mild but significant reduction in DAF-16 binding to the co-bound promoter regions in the absence of HLH-30. This indicates that HLH-30 mildly assists DAF-16 in binding to some promoters.

Regarding the comments by reviewer #1:

The majority of comments by reviewer #1 refer to one concern, namely that the reviewer is questioning whether DAF-16 and HLH-30 actually collaborate, are interdependent, and form a transcriptional regulatory module. We certainly respect the reviewer's opinion, but we find that this concern is unwarranted. We would like to emphasize that our original manuscript already provided a large portfolio of biochemical (co-IP, mRNA-seq, ChIP-seq) and genetic evidence (lifespan, oxidative stress resistance,...), all of which supported and was fully consistent with our claims. And in particular our genetic interaction data sometimes fully contradicted the reviewer's concerns. In the point-by-point rebuttal below, we now hope to alleviate the reviewer's concerns, by clarifying the implications of our original data and also referring to the new experimental evidence mentioned above, which should yet further strengthen and support our claims.

Comments by Reviewer #1:

In this work the authors start with a pulldown of DAF-16::GFP to identify binding partners (co-IP and mass spec) and in addition to known partners such as FTT-2, PAR-5, and SWI/SNF. HLH-30 bound to DAF-16 in daf-2 background (activated DAF-16, nuclear), and this was validated; results suggest that the interaction is not mediated by RNA or DNA (chromatin). HLH-30/TFEB is a well-known transcription factor that has been already described to be involved in lifespan regulation and autophagy. The translocation results suggest that the two TFs act independently under a wide range of conditions, eliminating binding to one another for the purpose of translocation as a mechanism.

Our response: While we agree with everything mentioned in the prior sentences, the last sentence is a bit misleading. Our translocation data only shows that the two transcription factors a) translocate to the nucleus under a wide range of conditions and b) that they don't depend on each other for this translocation. Our translocation results don't speak to the matter of whether the two transcription factors "act" together or independently for other purposes, most importantly whether they cooperate in the regulation of stress resistance and longevity promoting genes in the nucleus.

Loss of daf-16 and hlh-30 genetically suggests the two TFs are both independently required for daf-2 and glp-1 lifespan extension.

Our response: Here we unfortunately need to disagree with the reviewer. Yes, DAF-16 and HLH-30 are each individually required for the lifespan extension observed in *daf-2* and *glp-1* animals. However, they genetically don't act independently. As shown in Fig 3, the two transcription factors depend on each other for the promotion of these lifespan extension phenotypes. DAF-16 or HLH-30 have mostly no lifespan extending capability, if the other transcription factor is not present. Consistently, losing not just one but both of the transcription factors has hardly any additive effect on lifespan.

(I'm not sure how "epistatic" is being used in the text, as no order is established when both lof mutations lack an additive effect.)

Our response: We assume that the reviewer is referring to the sentence "..., suggesting that with regard to their genetic interaction DAF-16 and HLH-30 are epistatic." We carefully investigated this matter and believe that we made here acceptable use of the term "epistatic", as one can also read at the following link at Nature Education (<https://www.nature.com/scitable/topicpage/Epistasis-Gene-Interaction-and-Phenotype-Effects-460>): "Any time two different genes contribute to a single phenotype and their effects are not merely additive, those genes are said to be epistatic." Thus, to our understanding, no specific order needs to be established to call a gene interaction epistatic. Nevertheless, based on our genetic data, we actually think that *daf-16* and *hlh-30* function at the same level of the pathways promoting longevity and oxidative stress resistance, and there show "duplicate recessive epistasis". The term is explained at the following links:

<https://www.nature.com/scitable/topicpage/Epistasis-Gene-Interaction-and-Phenotype-Effects-460>,
<http://www.biologydiscussion.com/genetics/gene-interactions/top-6-types-of-epistasis-gene-interaction/37818>

We have now adjusted the sentence, hoping to improve clarity for the readers: “Second, we observed that combined loss of both transcription factors in *daf-16*; *hlh-30* mutant animals had hardly any additive effect (Fig. 3a-c), suggesting that *daf-16* and *hlh-30* are here in a relationship of duplicate recessive epistasis, functioning in the same genetic pathway.”

*Transcriptional analysis shows that many of DAF-16's targets are also HLH-30 targets, and chIP-seq experiments suggest that HLH-30 activates expression rather than repressing. Additionally, motif analyses suggest that simple promoter binding of the two TFs to their respective motifs (DBE and E-box) regulate binding, rather than the emergence of a new motif. Additionally, the binding of PQM-1 via the DAE may be involved in HLH-30 "baseline" expression. Dauer formation is surprisingly increased by loss of *hlh-30*, opposite to *daf-16*.*

-Tissue localization: if the tissue-specific DAF-16 expression lines (Libina, et al 2003; Zhang et al 2013) are used for co-IP/MS (intestine, neurons, muscle, hypodermis) is HLH-30 still pulled down in all of them? A more direct method of interaction should be used to address the tissue-specific interaction between HLH-30 and DAF-16, rather than relying on GFP results.

Our response: We appreciate the reviewer's interest in the tissue-specific aspects of our work, i.e. in which tissues the DAF-16–HLH-30 complexes are forming. Our microscopy data described co-expression and co-localization of DAF-16 and HLH-30 in most tissues of the organism, suggesting that also the interaction of DAF-16 and HLH-30 might occur broadly. However, our microscopy data is certainly no proof for it, nor are we making any such claims in the manuscript.

The reviewer now suggests to conduct tissue-specific co-IPs or related experiments, to more rigorously test a potential tissue-specificity of the interaction. However, while such co-IPs are theoretically doable, we saw the following concerns:

- 1) In our manuscript we provide robust data showing a physical interaction between DAF-16 and HLH-30, when investigating whole animals. We even further confirmed and characterized this interaction by our new in vitro binding assay, showing that the interaction between DAF-16 and HLH-30 is the result of a direct physical contact between the proteins (Fig. 1d). Furthermore, we now showed that complex formation of DAF-16/FOXO and HLH-30/TFEB transcription factors is conserved even in humans (Fig. S1c).

Taken together, this data should be sufficient to support the story told by our manuscript. Dissecting the DAF-16–HLH-30 interaction down to individual tissues would merely expand the already extensive data and information provided, but it wouldn't change the conclusions of our manuscript nor add immense value to it.

- 2) The proposed tissue-specific IP-mass spec experiments would have been challenging and difficult to complete within the provided revision timeframe. *C. elegans* has comparatively little intracellular protein, requiring such experiments to be conducted at very large scale.
- 3) If one would start to explore the tissue-specific aspects of our work biochemically, then one should probably also do it genetically, which would have meant yet additional experiments.

Taken together, we think that the tissue-specific exploration of the DAF-16–HLH-30 interaction would have gone beyond the scope of this manuscript and should rather be subject of future studies. We also discussed this matter with the editor, who agreed with us. We therefore hope for the reviewer's understanding that we decided to omit such experiments.

-Which tissue was used to score nuclear translocation?

Our response: The intestine. We apologize for not having clearly stated this in the manuscript and have now added this information to the results and methods sections.

-The translocation experiments, genetic evidence, transcription results, and chIP-seq data all suggest that the two TFs act independently. What is the evidence for this sentence?: "This is consistent with our hypothesis that DAF-16 and HLH-30 promote longevity by both of them"

getting activated, forming a complex, and jointly binding to promoter regions, where they promote expression of downstream genes important to prevent aging and thus extend the lifespan of the animal." or "We have established that DAF-16 tightly cooperates with HLH-30 in the promotion of longevity."

Our response: We appreciate the reviewer's comment, but unfortunately, we have to disagree with the reviewer's assessment that the two TFs would act simply independently. At least genetically, they are beyond doubt dependent on each other – for the purposes of promoting longevity or resistance to oxidative stress (Fig. 3 and Fig. 6a). Furthermore, no data in our manuscript contradicts the two sentences cited here by the reviewer. Instead we provide best possible evidence to support our claim of the two transcription factors forming a complex and often acting collaboratively, which (contrary to the reviewer's interpretation) we believe is strongly supported by our genetic and biochemical data:

1) Biochemical evidence:

- a. (co-)IP data: DAF-16 and HLH-30 form a complex (Fig. 1, S1). This physical interaction data is very solid; and we now even show that such complex also forms in humans (Fig. S1c). The fact that DAF-16 and HLH-30 form a complex is no proof but at least consistent with the two TFs collaborating. Furthermore, from an evolutionary perspective, there would be little sense in having/evolving a physical interaction between two TFs, if there was not some mechanistic purpose to it, i.e. mechanistic cross-talk or synergy between the TFs.

To yet further support this notion, we now provide also *in vitro* binding data, showing that the two transcription factors have evolved the ability to bind to each other even in the absence of any other protein, DNA, or RNA (Fig. 1d).

- b. Size-exclusion chromatography: In Figure 1e, we now show our new size-exclusion chromatography data, in which we observe that DAF-16- and HLH-30-containing complexes co-migrate in identical fractions and show identical shifts to yet higher molecular weights – specifically under conditions that require them to act in the same genetic pathway, namely under low IIS or oxidative stress but not heat stress. This data would be inconsistent with two independently acting TFs. Why would they strictly co-migrate, if they would not co-reside and incorporate into the same protein complexes? Why do we observe perfectly synchronized shifts of both TFs to the same higher weight fractions, if they would not be regulated or acting together?
 - c. ChIP-seq: If two transcription factors would act independently, only coincidentally co-regulating some target genes, one would expect no significant overlap between their binding sites on chromatin, and these sites should be randomly spaced. However, if two transcription factors reside in a complex and presumably cooperate in gene regulation, one would expect them to reside not only in some of the same promoter regions but in identical locations. Consistently, Fig. 5c shows that 41% of DAF-16 peaks overlap with HLH-30 peaks – an exceptionally high overlap that cannot arise by chance ($p < 10^{-200}$). Further, the distance between DAF-16 summits and HLH-30 summits is preferentially zero, as illustrated in Fig. S3. Finally, we now show in Figure 5f that there is a mild synergy between the two TFs in binding to DNA. In the absence of HLH-30, DAF-16 binding to co-bound promoter regions is mildly but significantly reduced.
 - d. mRNA-seq: If two transcription factors would act independently, only coincidentally co-regulating some target genes, one would expect no significant overlap between their target gene sets. However, DAF-16 and HLH-30 significantly overlap in the genes they regulate, with p-values ranging from 10^{-37} to 10^{-191} . Hundreds of genes require the presence of both transcription factors to reach their normal expression levels – a compelling example of their synergy (Fig. 4c-e and 7c,d).
- 2) Genetic evidence: If DAF-16 and HLH-30 would act independently of each other, one would expect that each of them is sufficient to confer stress resistance and longevity, no matter whether the other transcription factor is present. Combined loss of both transcription factors should lead to fully additive phenotypes. However, we clearly show in Figures 3 and 6a that for the promotion of longevity and oxidative stress resistance the two TFs require each other and as a consequence do not act additively. This can only be explained by DAF-16 and HLH-30 residing

in the same genetic pathway and is incompatible with the reviewer's hypothesis of independent actions – at least from the genetic perspective.

Nevertheless, there are indeed times where the two TFs act independently, but only for conferral of heat stress resistance and for induction of dauer formation, which we illustrate in Figure 6b-d.

Taking all this together, we hope to have provided sufficient evidence against the reviewer's hypothesis of DAF-16 and HLH-30 acting strictly independently, but rather have provided a broad spectrum of data in support of DAF-16 and HLH-30 forming a complex and cooperating in the regulation of many target genes, at least when promoting phenotypes like oxidative stress resistance and longevity.

How would their model be different if HLH-30 had simply been characterized without co-IP data? No cooperation is established by any of the data presented to this point in the paper. The transcriptional data and chIP-seq data do not point to the formation of a complex, but rather the independent regulation of two stress-sensitive TFs. Cooperation would be shown by synergetic interactions, which is not shown here; instead, all of these data besides co-IP paint a picture of two ships passing in the night in response to the same beacon but totally unaware of one another.

Our response: First, we don't think it is very reasonable to ask, if a manuscript is still coherent with a key experiment (like here the co-IPs) taken out of it. Scientific publications often rely on the entirety of their experiments to arrive at the conclusions that they present. In the case of our manuscript, the (co-)IP data provides indeed very strong and important evidence for the DAF-16–HLH-30 complex formation and cooperation, and we would not want to miss this data. However, we now newly provide also in vitro binding data that independently confirm the ability of DAF-16 and HLH-30 to form complexes (Fig. 1d). Further, our Chip-seq data points to the formation of a complex, given that the two TFs co-localize at such high frequency to the same chromosomal positions (Fig. 5c).

The reviewer further states: “No cooperation is established by any of the data presented to this point in the paper.” and “Cooperation would be shown by synergetic interactions, which is not shown here”. Here we again would like to disagree with the reviewer: We show exactly such cooperation or synergy, e.g. by our genetic and mRNA-seq data: Both transcription factors need to synergize to promote longevity (Fig. 3), oxidative stress resistance (Fig. 6a) and the appropriate expression of many target genes (Fig. 4, 7). If only one of the two TFs is missing, this all fails and the second transcription factor becomes ineffective. We kindly would like to argue that this is rather described by “two ships sailing in the dark in response to the same beacon, where one ship has only a map and the other only a compass, so that they have to synergize and combine their resources to find their correct path and destination”. As final evidence for synergy between the two TFs, we would like to again mention the data presented in Figure 5f, which shows that HLH-30 mildly aids DAF-16 in binding to co-bound promoter regions.

The dauer results further support this view, as DAF-16 and HLH-30 have opposite roles.

Our response: As we show in the manuscript, heat shock resistance and dauer formation are not conferred by genes co-regulated by DAF-16 and HLH-30. Thus our dauer results are not contradicting that DAF-16 and HLH-30 cooperate in other contexts like longevity promotion or ox stress responses.

The authors discuss in the suggestion that these data shift the paradigm of the field, but there is no data shown here that support the notion that these two TFs form a transcriptional regulatory module; the data suggest that HLH-30 regulates a subset of DAF-16's targets--independently of DAF-16--, that it is often found on the promoters of DAF-16 target genes, and that most (but not all) of the time, HLH-30 also responds in the same way to environmental conditions as DAF-16 does.

Our response: First, we are convinced that our work is very important for the aging research community and will have strong influence on future work in this field. This is also the feedback that we have gotten at conferences, and it is further illustrated by some of the new data that we now provide in this revision. Most importantly, our data showing that complex formation between DAF-16/FOXO and HLH-30/TFEB is conserved in humans will almost certainly lead to exciting follow up studies.

Second, our use of the term “transcriptional regulatory module” to describe the cooperation between DAF-16 and HLH-30 should be absolutely appropriate. Our genetic data shows beyond doubt that DAF-16 and HLH-30 cooperate in the promotion of longevity and oxidative stress resistance phenotypes. And our data shows beyond doubt that the two TFs function as combinatorial transcription factors, because they bind to many of the same promoter regions and regulate overlapping gene sets. This genetic interaction and combinatorial role in gene regulation alone should already suffice to use the term “transcriptional regulatory module” – even if the two TFs would not act as a complex nor physically interact.

How would the authors distinguish their model from one in which two TFs that function independently but are nuclear localized under similar conditions end up near each other on promoters under stress conditions, but do not directly or synergetically act, but instead are simply more often near each other, so under cross-linking conditions they are more likely to bind to one another due to proximity rather than a real complex formation?

Our response: Our data shows that the alternative model proposed by the reviewer is clearly wrong, for two reasons:

First, because our two TFs do synergize. Our genetic data clearly shows that they require each other to confer longevity and ox stress resistance. Furthermore, we now have provided new evidence in Figure 5f whereby DAF-16 and HLH-30 show partial synergy in promoter binding, with HLH-30 mildly but significantly aiding DAF-16 in binding to co-occupied promoter regions.

Second, because we did not cross-link for our large-scale IPs, our co-IPs, nor for our new in vitro binding assays. And in some of the co-IPs and in the in vitro binding assays DNA and RNA were entirely absent due to treatment with nucleases. Given that we still identified a physical interaction between DAF-16 and HLH-30 in at least five independent experiments illustrates that their interaction is absolutely not a result of coincidental proximity. DAF-16 and HLH-30 truly can bind to each other.

It seems that this is in fact the more likely situation; otherwise the stoichiometry should be 1:1, and there should be evidence of cooperatively rather than independence.

Our response: Please see all our comments above in support of the DAF-16–HLH-30 complex and the synergy of these transcription factors in combinatorial gene regulation.

Furthermore, we don't show any data that argues for or against the two TFs binding to each other in a 1:1 stoichiometry. Different types of experiments would be needed. It is unclear to us, how the reviewer can conclude from the provided data that the stoichiometry is not 1:1.

Overall, this manuscript does a good job of describing all of the myriad ways that HLH-30 acts similarly to DAF-16 and presents a great deal of information about HLH-30 activity, but the main point, that it binds to DAF-16, remains almost entirely unexplored beyond the initial co-IP results; the nature of this physical interaction that initiated the study is not rigorously explored. While that is not critical for the basic description of HLH-30's targets under various conditions (which itself is valuable information), the use of words like "cooperative" and "complex" to describe its interactions are not justified by the data shown here.

Our response: We appreciate the reviewer's feedback and are glad that the reviewer sees the value of our gene expression analyses and our comparative exploration of DAF-16 and HLH-30 functions. However, we are convinced that also our evidence for complex formation and cooperation between DAF-16 and HLH-30 is reliable:

The binding and therefore complex formation between the two TFs was shown not just by one but by three different IP approaches already in the original manuscript. Thus, we should be allowed to use the term “complex”.

Nevertheless, we followed the reviewer's request and explored the binding between the two TFs by yet additional experiments – all mentioned already above:

- 1) We now showed by new co-IPs that the complex formation is also conserved in humans – in particular an interaction between TFEB and FOXO1 (Fig. S1c),

- 2) We now showed by in vitro binding assays that the interaction between DAF-16 and HLH-30 is direct and not mediated by other proteins or DNA/RNA.
- 3) We conducted size-exclusion chromatography (Fig. 1e, S1b), showing strict co-migration between DAF-16 and HLH-30. Furthermore, we observed their concerted shift to higher molecular weights under stimuli that elicit their cooperation in the same genetic pathway, i.e. low IIS and oxidative stress. Given that we had already shown that the interaction between the two TFs is becoming more abundant under low IIS, this data together suggests that the interaction between DAF-16 and HLH-30 may actually be promoted by certain upstream stimuli. Promotion of the interaction would in turn promote the expression of co-regulated target genes. It will be interesting to explore this regulation and its mechanistic basis in the future.

Finally, we believe that our use of the term “cooperative” is not unreasonable, too. Possibly the reviewer prefers a different interpretation of this term, but in the end there should be many different means of cooperation, and we have surely laid out a broad variety of evidence in this rebuttal that describes how the two transcription factors are not blind to each other but interact – physically and genetically – and cooperate as part of a sophisticated transcriptional regulatory module to promote stress resistance and longevity.

To further support our case, we also surveyed the literature and found the term “cooperative” being used in many instances, where the interplay between two transcription factors is less direct and the experimental evidence for cooperation is much weaker than in our case of the DAF-16–HLH-30 cooperation. Notably, this term is used, even if there is no physical interaction between the TFs but only a regulation of overlapping target gene sets occurs. Here two examples, related to DAF-16, where the other transcription factor does not bind to DAF-16 but merely co-regulates some target genes:

- 1) “...several studies have shown that the *C. elegans* HSF-1 cooperates with DAF-16 to increase lifespan through reduced DAF-2 signaling...”, published in².
- 2) “Given that DAF-16 and SKN-1 are inhibited in parallel by IIS, and cooperate to regulate some target genes...”, published in³.

Comments by Reviewer #2:

The Lin et al manuscript reports an interesting collaboration between two transcription factors DAF-16 and HLH-30 in longevity and stress response. Whereas both DAF-16 and HLH-30 have been shown to be important for longevity, their working relationship was unknown. The authors revealed a partnership between the two factors via IP mass spec. Using further genetic and genomic analyses, they presented data to support that DAF-16 and HLH-30 form a protein complex and are recruited to the common set of promoters to regulate their gene expression. More interestingly, under some conditions, DAF-16 and HLH-30 each have their own regulatory targets and thus different functional outcomes. Therefore, DAF-16 and HLH-30 both act together and separately to regulate various physiological outputs.

In general, the data presented are of good quality and largely support the conclusions.

A few comments:

For Fig. 1, representative western blot and nuclear localization results should be shown. The bar graphs are effective for summarizing the results, but actual images that represent the different results will give a better idea about what the results actually look like.

Our response: Regarding representative western blots: Figures 1a and 1b were only analyzed by mass spectrometry. We cannot provide western blot data demonstrating the DAF-16–HLH-30 interaction in these large-scale IPs, since HLH-30 is only expressed endogenously and we don’t have an antibody that detects it. However, we conducted a co-IP using DAF-16::Flag and HLH-30::GFP co-expressing worms followed by western blotting to confirm the observed interaction and show a representative western blot in Fig. 1c. Thus, we feel that we provided representative western blot data wherever it was possible and appropriate. Furthermore, by in vitro binding and gel filtration experiments presented in Figures 1d,e

and S1b,c we now provide yet additional evidence to support and characterize the physical interaction between DAF-16 and HLH-30 – all of which is documented by western blotting, too.

Regarding representative images of the nuclear localization data: Some examples of nuclear localization images are shown in Fig. 2a and 2b. However, we fear that showing additional images for all the different stimuli and their resulting different degrees of nuclear translocation would lead to a Figure of unreasonable size. Nevertheless, to address the reviewer's request for representative images, we now added Figure S2a that shows representative images of the different categories of nuclear localization of DAF-16 and HLH-30 that were scored in Figures 2c and S2b.

In the Method section, it was stated that all experiments were done at least as duplicates. However, based on the data presented, it is difficult to know how consistent the replicates are. For lifespan data, the quantitative data of the individual trials should be presented as supplementary data.

Our response: We apologize for omitting the statistical information on the replicate lifespan and stress survival experiments. We now added this information to Table S2.

For the RNA-seq and ChIP-seq data, the overall correlation of the data between the replicates should be shown, e.g. via a simple scatter plot or correlation or PCA analysis. Based on the Method, it is also not clear how the duplicate data are processed. Are the data simply pooled, or are the data analyzed as separate replicates and does the data processing pipeline take the replicates into account and only report consistent peaks?

Our response: For the various mRNA-seq and ChIP-seq experiments, we now provide information on the correlation of the different biological replicates in Figure S8.

Of our mRNA-seq conditions, some conditions were analyzed in duplicate, some in triplicate, depending on the experiment and the expected level of noise. Pearson correlations between replicates were usually very high, above 0.9, with only very few outliers. Our mRNA-seq processing pipeline, based on Tophat 2 and Cuffdiff 2, took the different replicates into account. Thus, the presented gene expression data is based on all replicates. This information is now provided in the mRNA-seq methods section.

Our ChIP-seq data processing pipeline did not take replicates into account. The ChIP-seq data presented is based on an individual experiment. However, we reproduced our ChIP-seq data in independent replicate experiments, confirming reproducibility by observing very good Pearson correlations between the replicate datasets of at least 0.927. The respective methods section has now been updated to provide this information.

In summary, we once again would like to thank the reviewers for their constructive feedback. We considered all their comments very carefully and tried to address them either experimentally or in writing. We sincerely believe that by this revision our manuscript has now substantially improved, that it reached a high standard, and we hope that it finds the support of the reviewers, too.

References:

1. Riedel, C. G. *et al.* DAF-16 employs the chromatin remodeller SWI/SNF to promote stress resistance and longevity. *Nat. Cell Biol.* **15**, 491–501 (2013).
2. Anckar, J. & Sistonen, L. Regulation of HSF1 function in the heat stress response: implications in aging and disease. *Annu. Rev. Biochem.* **80**, 1089–115 (2011).
3. Wang, J. *et al.* RNAi screening implicates a SKN-1-dependent transcriptional response in stress resistance and longevity deriving from translation inhibition. *PLoS Genet.* **6**, (2010).

REVIEWERS' COMMENTS:

Reviewer #1 (Remarks to the Author):

I am satisfied with the explanations and new data provided by the authors to support their work.

Reviewer #2 (Remarks to the Author):

The authors satisfactorily addressed my previous comments. The data presented are of high quality and the paper will be an important addition to the field.

REVIEWERS' COMMENTS:

Reviewer #1 (Remarks to the Author):

I am satisfied with the explanations and new data provided by the authors to support their work.

Our response: We are happy that the reviewer is satisfied with the revised manuscript.

Reviewer #2 (Remarks to the Author):

The authors satisfactorily addressed my previous comments. The data presented are of high quality and the paper will be an important addition to the field.

Our response: We are happy that the reviewer is satisfied with the revised manuscript.